# A single residue in the yellow fever virus envelope protein modulates virion architecture and antigenicity

Summa Bibby [1,8], James Jung [1,6,8], Yu Shang Low [1], Alberto A. Amarilla [1], Natalee D. Newton [1,2], Connor A. P. Scott [1], Jessica Balk [3], Yi Tian Ting[3,7], Morgan E. Freney[1], Benjamin Liang [1], Timothy Grant [4,5], Fasséli Coulibaly [3], Paul Young[1,2], Roy A. Hall[1,2], Jody Hobson-Peters [1,2], Naphak Modhiran [1,2,8] ✉ & Daniel Watterson [1,2,8] ✉

Yellow fever virus (YFV) is a re-emerging flavivirus that causes severe hepatic disease and mortality in humans. Despite being researched for over a century, the structure of YFV has remained elusive. Here we use a chimeric virus platform to resolve the first high resolution cryo-EM structures of YFV. Stark differences in particle morphology and homogeneity are observed between vaccine and virulent strains of YFV, and these are found to have significant implications on antibody recognition and neutralisation. We identify a single residue (R380) in the YFV$_{17D}$ envelope protein that stabilises the virion surface, and leads to reduced exposure of the cross-reactive fusion loop epitope. The differences in virion morphology between YFV strains also contribute to the reduced sensitivity of the virulent YFV virions to vaccine-induced antibodies. These findings have significant implications for YFV biology, vaccinology and structure-based flavivirus antigen design.

Yellow fever virus (YFV), the etiological agent of yellow fever (YF), is endemic to the tropical and sub-tropical regions of Africa and South America[1]. YF can vary in humans from a mild flu-like illness to a fulminating, fatal disease[2]. YFV primarily circulates in a sylvatic cycle involving canopy-breeding mosquitoes and non-human primates. However, human encroachment into forested areas can lead to sporadic infections when individuals are bitten by infected sylvatic mosquitoes. Humans can then serve as viremic hosts, contributing to the urban transmission of yellow fever, with *Aedes aegypti* playing a significant role in facilitating this transmission[1]. YFV outbreaks have periodically occurred in endemic regions since the 1960s; however, in recent years, outbreaks have been recorded in both endemic and non-endemic areas[3–7]. During the 2016-2019 YFV epidemic in south-eastern Brazil, the virus spread into several urban states that had not seen YFV transmission in over 90 years[8,9]. As no therapeutic agents are approved for treating YFV infections, the primary method of controlling YFV outbreaks is through mass vaccination campaigns with the live-attenuated YFV$_{17D}$ vaccine[10].

The YFV$_{17D}$ vaccine strain was developed in 1937 by serially passaging the virulent YFV$_{Asibi}$ strain in mouse and chick embryo tissues[11]. Further passaging of the original vaccine strain yielded the three phenotypically indistinguishable sub-strains currently used today: 17D-

[1]School of Chemistry & Molecular Biosciences, The University of Queensland, St Lucia, QLD, Australia. [2]Australian Infectious Disease Research Centre, The University of Queensland, St Lucia, QLD, Australia. [3]Department of Biochemistry and Molecular Biology and Biomedicine Discovery Institute, Monash University, Clayton, VIC, Australia. [4]John and Jeanne Rowe Center for Research in Virology, Morgridge Institute for Research, Madison, WI, USA. [5]Department of Biochemistry, University of Wisconsin-Madison, Madison, WI, USA. [6]Present address: Janelia Research Campus, Howard Hughes Medical Institute, Ashburn, VA, USA. [7]Present address: Centre for Inflammatory Diseases, Department of Medicine, School of Clinical Sciences, Monash University, Clayton, VIC, Australia. [8]These authors contributed equally: Summa Bibby, James Jung, Naphak Modhiran, Daniel Watterson. ✉e-mail: n.modhiran@uq.edu.au; d.watterson@uq.edu.au

204, 17D-213 and 17DD[12]. The YFV$_{17D}$ vaccine elicits a complex, integrative and multilineage immune response in vaccinees and confers protection against virulent YFV strains through a potent and long-lasting neutralising antibody response[13–18]. Whilst it has always been thought that the YFV$_{17D}$ vaccine protects against all known wild-type strains of the virus, recent findings have found that an emergent Brazilian strain of YFV (YFV$_{ES504}$) has reduced susceptibility to vaccine-induced antibodies[19,20]. This reduced susceptibility was genetically mapped to two sites on the YFV envelope (E) protein that are shared amongst most South American strains, suggesting that the YFV$_{17D}$ vaccine is non-optimal for the current circulating strains[20].

Resolving the antigenic landscape of the YFV virion is key to understanding this observed differential neutralisation by vaccine-induced antibodies, as well as any future structure-based vaccine design. However, despite successful elucidation of many other flavivirus virions by cryogenic electron microscopy (cryo-EM), no whole virion structures of the infectious mature YFV particle are available[21–25]. Crystallographic studies have resolved the structure of the YFV E protein and shown that its domain architecture is similar to related flaviviruses, with three N-terminal ectodomains, known as domains I (DI), II (DII) and III (DIII) and a stem anchor region known as the transmembrane domain (TM)[26–28]. Besides these crystal structures, the only other YFV structure currently available is a low resolution cryo-EM reconstruction of the immature YFV$_{17D}$ virion[29]. Given that many potently neutralising antibodies described for related flaviviruses recognise quaternary sites on the mature virion, there is a pressing need to better understand the mature YFV virion structure[30–32].

To investigate the structure of both vaccine and pathogenic strains of YFV, we utilised the well-established Binjari virus (BinJV) platform as it has a robust safety profile and accurately recapitulates the virion structure and immunological landscape of pathogenic flaviviruses[33,34]. Chimeric viruses consisting of the pre-membrane (prM) and E structural genes of either YFV$_{17D-204}$ (vaccine strain), YFV$_{ES504}$ (recently circulating virulent strain) or YFV$_{Asibi}$ (parental virulent strain), and the genomic backbone of BinJV were generated via a stream-lined amplicon based method[33,35]. Here we present cryo-EM structures of these chimeric YFV virions, revealing remarkable differences at the virion level. We also suggest a molecular basis for YFV mature virion dynamics and concomitant exposure of the cryptic fusion loop epitope (FLE). These findings have broad implications for YFV biology and serve as valuable structural templates for structure-based flavivirus vaccine design.

## Results

### YFV$_{17D}$ is structurally and antigenically different to YFV$_{ES504}$ and YFV$_{Asibi}$

To resolve the virion structure of vaccine and virulent strains of YFV via cryo-EM, three BinJV-YFV$_{prME}$ (bYFV) chimeras were generated and gradient purified: bYFV$_{17D}$, bYFV$_{ES504}$ and bYFV$_{Asibi}$ (Supplementary Fig. 1a). The viruses were assessed for purity via sodium dodecyl sulphate-polyacrylamide gel electrophoresis (SDS-PAGE) (Supplementary Fig. 1b) and imaged using cryo-EM (Fig. 1a & Supplementary Fig. 1c). Whilst the bYFV$_{17D}$ particles appear smooth like typical mature flavivirus particles, the bYFV$_{ES504}$ and bYFV$_{Asibi}$ virions exhibit bumpy and uneven surfaces (Fig. 1a). To further compare the structure of vaccine and virulent YFV virions, large cryo-EM data sets of bYFV$_{17D}$ and bYFV$_{ES504}$ were captured. The observed structural differences between the vaccine and virulent strain virions were further emphasised in the 2D class averages generated by single particle analysis (SPA) (Fig. 1b). 3D reconstructions of bYFV$_{17D}$ and bYFV$_{ES504}$ were resolved by SPA to resolutions of 7.5 Å and 12.6 Å, respectively (Fig. 1c-d, Supplementary Fig. 2, Supplementary Fig. 3 and Supplementary Table 1). The cryo-EM structure of bYFV$_{17D}$ displays the typical flavivirus architecture of 180 E proteins arranged as rafts of three head-to-tail dimers that form a herringbone-like pattern on the surface of the

icosahedral virions. Clear densities corresponding to the transmembrane domains of E as well as the membrane protein (M) are also observed in the bYFV$_{17D}$ reconstructions (Fig. 1c), although these are at a lower resolution relative to the E ectodomain (Supplementary Fig. 3). In contrast, the cryo-EM density map of bYFV$_{ES504}$ lacks any typical mature flavivirus characteristics (Fig. 1d). The central cross-section shows no clear definition between the E protein layer and the lipid bilayer (Fig. 1d).

To explore the possible effects that these observed structural differences may have on the YFV antigenic landscape, a panel of anti-YFV and anti-flavivirus monoclonal antibodies (mAbs) were generated as recombinant hIgG1 mAbs and tested for binding and neutralisation against bYFV$_{17D}$, bYFV$_{ES504}$ and bYFV$_{Asibi}$[26,36–40]. The anti-YFV mAbs 5A and 2C9 (DII binders) and the anti-flavivirus mAbs 2A10G6 and 6B6C-1 (FLE binders) bound to purified particles of bYFV$_{17D}$, bYFV$_{ES504}$ and bYFV$_{Asibi}$ (Supplementary Fig. 4). As expected, the YFV$_{17D}$-specific mAb, 864 (DIII binder) only bound to bYFV$_{17D}$ (Supplementary Fig. 4a). The in vitro neutralisation activity of these mAbs against the bYFVs was then assessed using focus-reduction neutralisation tests (FRNTs). 5A and 2C9 neutralised all three bYFVs. However, 2A10G6 and 6B6C-1 only displayed neutralisation activity against bYFV$_{ES504}$ and bYFV$_{Asibi}$ (Fig. 1e). As expected, mAb 864 only neutralised bYFV$_{17D}$ (Fig. 1e). To further tease apart the unique neutralisation profile of the fusion-loop mAbs against the bYFVs, both bYFV$_{17D}$ and bYFV$_{ES504}$ were complexed with 2A10G6 Fab and imaged via cryo-EM (Supplementary Fig. 4b). 2A10G6 Fab complexes well with bYFV$_{ES504}$ as the virion surfaces appear to be completely covered in Fab-like densities. Interestingly, 2A10G6 Fab appears to bind poorly to bYFV$_{17D}$, as only a few Fab-like densities can be observed on the surface of the bYFV$_{17D}$ virions (Supplementary Fig. 4b). Next, the neutralising antibody response to YFV$_{17D}$ vaccination in 14 human donors was assessed against bYFV$_{17D}$, bYFV$_{ES504}$ and bYFV$_{Asibi}$ (Fig. 1f and Supplementary Table 2). Comparison of the serum neutralising titres showed a significant decrease in neutralisation of both bYFV$_{ES504}$ and bYFV$_{Asibi}$ when compared to bYFV$_{17D}$ (Fig. 1e). Together, these results suggest structural and antigenic differences between vaccine and virulent strains of YFV.

### Fab complexing reveals high-resolution structure of bYFV$_{17D}$

To further explore the structural differences between vaccine and virulent strains of YFV, bYFV$_{17D}$ and bYFV$_{ES504}$ were independently complexed with 5A and 2C9 Fabs at 4 °C and imaged using cryo-EM (Fig. 2). SPA yielded cryo-EM maps of bYFV$_{17D}$:5A, bYFV$_{17D}$:2C9, bYFV$_{ES504}$:5A and bYFV$_{ES504}$:2C9 to resolutions of 11.8 Å, 7.3 Å, 21.8 Å and 15.5 Å, respectively (Fig. 2b, Supplementary Figs. 5–7 and Supplementary Table 1). The cryo-EM maps of bYFV$_{17D}$:5A and bYFV$_{17D}$:2C9 show Fabs occupying all 180 sites on the virion surface. In contrast, the density maps of bYFV$_{ES504}$:5A and bYFV$_{ES504}$:2C9 show incomplete and ill-defined Fabs on the virion surface (Fig. 2b), despite clear Fab complexation in the cryo-EM micrographs (Fig. 2a).

Next, we carried out symmetry expansion in cisTEM2 to obtain a 4 Å reconstruction of the bYFV$_{17D}$ asymmetric unit (ASU) complexed with Fab 2C9 (Fig.2c, Supplementary Fig. 8 and Supplementary Table 3)[41]. This 4 Å cryo-EM map of bYFV$_{17D}$:2C9 was used to build an atomic model of the ASU (Fig. 2c, d). We also determined a crystal structure of the free 2C9 Fab, which closely matched the cryo-EM structure (Supplementary Fig. 9 and Supplementary Table 4). Consistent with prior escape mutant data, this model identified seven residues (H67, K69, D72, C74, N86, E-87 and R99) in E-DII that contribute to 2C9 binding (Supplementary Fig. 9)[42]. Next, the cryo-EM structure of bYFV$_{17D}$ was compared with the X-ray crystal structure of YFV$_{17D}$ (Fig. 2e)[26]. As expected, the crystal structure lacks the native E-protein curvature, however the structures are highly similar overall, with an all atom RMSD of 1.2 Å (average from rigid fit of individual domains).

 

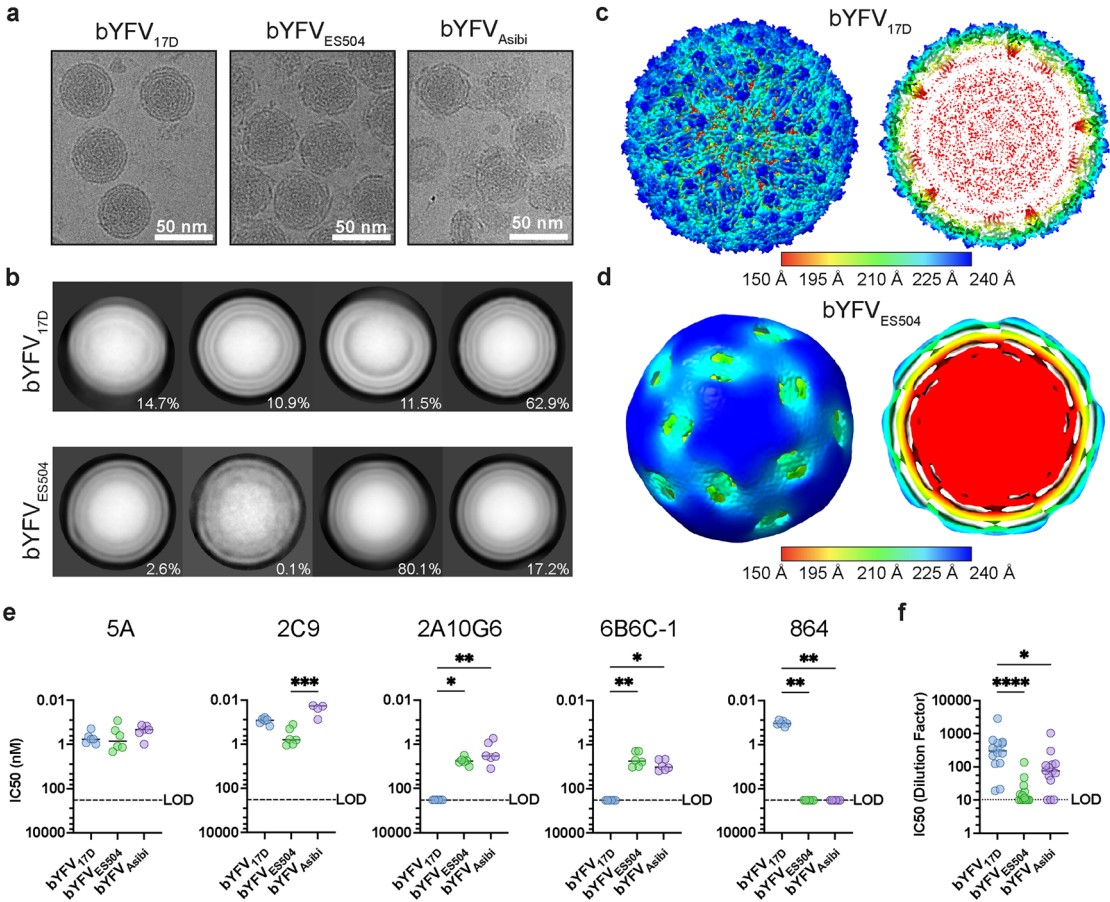

**Fig. 1 | Structural and antigenic characterisation of bYFVs. a** Representative cryo-EM micrographs of purified bYFV$_{17D}$, bYFV$_{ES504}$ and bYFV$_{Asibi}$ particles incubated at 4 °C prior to freezing. **b** Two-dimensional class averages of bYFV$_{17D}$ and bYFV$_{ES504}$ particles. The displayed class distribution is shown as a percentage of the total number of particles. Cryo-EM density map of bYFV$_{17D}$ (**c**) and bYFV$_{ES504}$ (**d**) with I3 symmetry applied. The maps are radially coloured according to the following: 0-150 Å red, 151-195 Å yellow, 196-210 Å green, 211-225 Å cyan, 226-240 Å blue Å. **e** IC50 values of recombinant hIgG1 anti-YFV or anti-flavivirus mAbs against bYFV$_{17D}$, bYFV$_{ES504}$ and bYFV$_{Asibi}$. Each symbol represents a technical replicate from three biological replicates ($n = 6$). **f** Serum neutralising titres of human YFV$_{17D}$

vaccinees ($n = 14$) against bYFV$_{17D}$, bYFV$_{ES504}$ and bYFV$_{Asibi}$. In both (**e**) and (**f**), neutralisation was determined via FRNTs on C6/36 (*Aedes albopictus*) cells. Lines indicate group medians. Significance was determined via Kruskal-Wallis tests with Dunn's multiple comparisons test on GraphPad Prism 9.0. ****$p < 0.0001$, ***$p \leq 0.0002$, **$p \leq 0.002$, *$p \leq 0.03$. For 2C9: $p = 0.006$ (ES504 vs. Asibi), 2A10G6: $p = 0.040$ (17D vs. ES504) and $p = 0.002$ (17D vs. Asibi), 6B6C−1: $p = 0.034$ (17D vs. ES504) and $p = 0.002$ (17D vs. Asibi), 864: $p = 0.0015$ (17D vs. ES504 and 17D vs. Asibi), human vaccinee sera: $p = 0.0001$ (17D vs ES504) and $p = 0.046$ (17D vs Asibi). LOD = limit of detection. Source data is provided as a Source Data file.

## DIII is a key determinant of YFV structure and antigenicity

With evidence of structural and antigenic differences between bYFV$_{17D}$ and bYFV$_{ES504}$, a panel of bYFV prME chimeras were generated to investigate the origin of these differences. Firstly, prM was swapped between the two strains, followed by each individual domain of E, producing a panel of 20 bYFV prME chimeras (Fig. 3a and Supplementary Fig. 10). Highly pure mature or partially mature preparations of each virus were obtained via gradient purifications (Supplementary Fig. 11). Immediately upon purification, each sequence confirmed bYFV chimera was imaged using cryo-EM (Fig. 3b). Interestingly, all the chimeric virions in the bYFV$_{17D}$ backbone retain their parental virus morphology, except for bYFV$_{17D}$/DIII$_{ES504}$, which more closely resembles the bumpy bYFV$_{ES504}$ phenotype. Similarly, all the chimeric virions in the bYFV$_{ES504}$ backbone retained their parental virus phenotype, except for bYFV$_{ES504}$/DIII$_{17D}$, which most closely resembles the smooth phenotype of bYFV$_{17D}$. These results suggest that E-DIII is playing a major role in YFV virion architecture.

To determine if E-DIII is contributing to the antigenic differences observed in Fig. 1e, we assessed the neutralisation activity of 5A, 2A10G6, 6B6C-1 and 864 against the bYFV DIII chimeras. Swapping DIII between bYFV$_{17D}$ and bYFV$_{ES504}$ had no effect on the neutralisation activity of 5A, however, it did have a significant effect on the

neutralisation activity of 2A10G6 and 6B6C-1 (Fig. 3c). The addition of DIII$_{ES504}$ in bYFV$_{17D}$ increased the neutralisation activity of both 2A10G6 and 6B6C-1, whilst the addition of DIII$_{17D}$ in bYFV$_{ES504}$ reduced the neutralisation activity of both 2A10G6 and 6B6C-1 (Fig. 3c). Acting as a DIII$_{17D}$ control, 864's neutralisation profile was significantly affected due to the exchange of DIII between bYFV$_{17D}$ and bYFV$_{ES504}$ (Fig. 3c). These findings indicate that DIII is contributing to the antigenic differences and these changes are correlated with virion morphology.

Next, a cryo-EM map of bYFV$_{ES504}$/DIII$_{17D}$ was resolved to a resolution of 6.5 Å (Fig. 4a, Supplementary Fig. 12, 13 and Supplementary Table 5). This reconstruction displays the typical flavivirus architecture of 180 E proteins arranged in a herringbone-like pattern on the virion surface (Fig. 4a) and is substantially better resolved than the bYFV$_{ES504}$ reconstruction (Figs. 1d and 4a). We then utilised the symmetry expanded local reconstruction method to resolve the ASU of bYFV$_{ES504}$/DIII$_{17D}$ to 3.3 Å (Fig. 4b, Supplementary Fig. 13 and Supplementary Table 3). Similar to the bYFV$_{17D}$:2C9 ASU reconstruction, M as well as the transmembrane domains of E have not been resolved. This would suggest that unlike for other classically flat or closed flavivirus virions such as ZIKV and DENV[22,43], the membrane-proximal regions are flexible and do not align well with the E ectodomain shell. Next, the

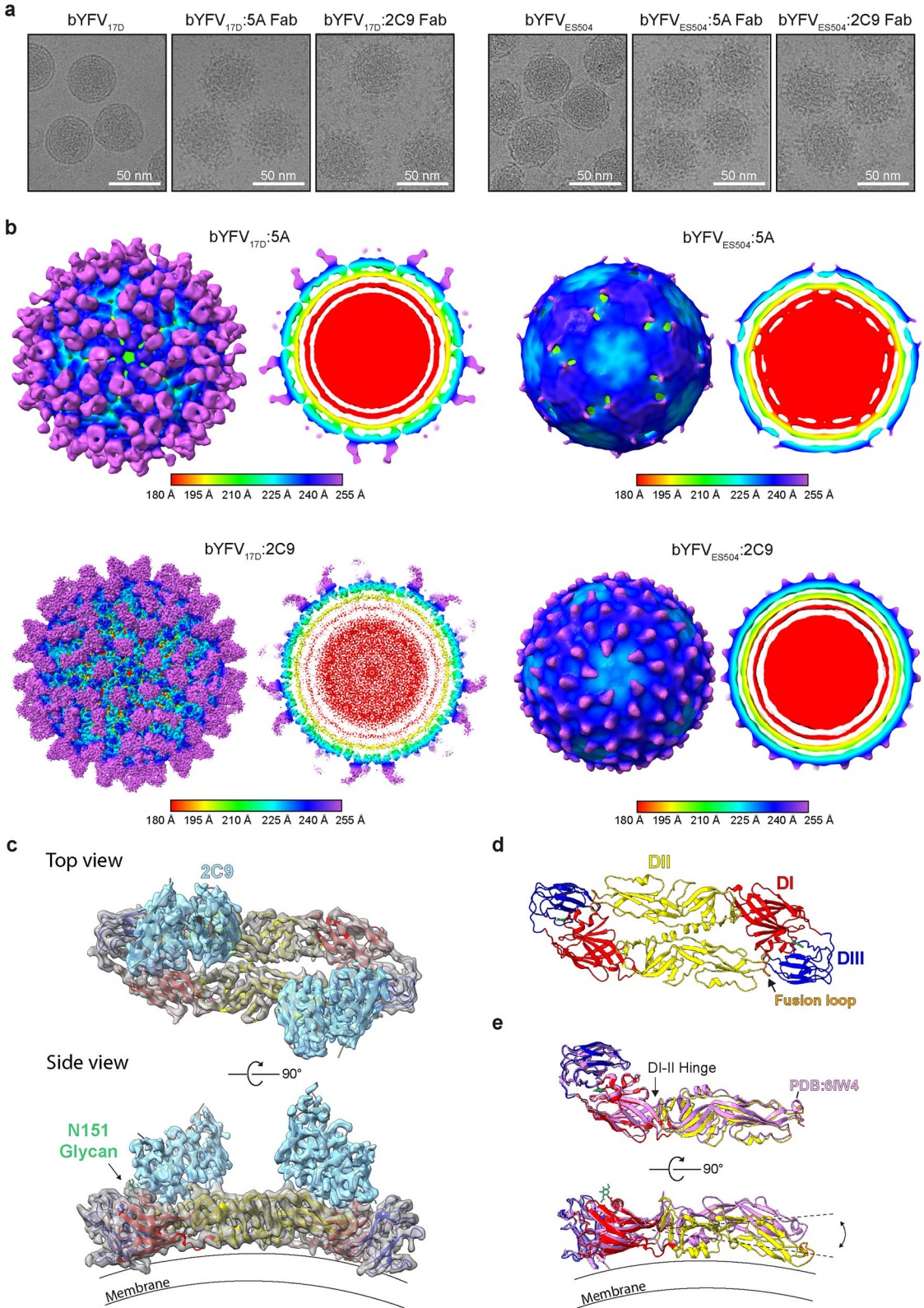

**Fig. 2 | Cryo-EM of bYFV$_{17D}$ and bYFV$_{ES504}$ complexed with 5A and 2C9 Fabs.**
**a** Representative micrographs of bYFV$_{17D}$ and bYFV$_{ES504}$ with and without complexing with 5A or 2C9 Fab. Viruses and Fabs were combined in a 1:1 ratio and incubated overnight at 4 °C prior to vitrification. **b** Cryo-EM density maps of bYFV$_{17D}$ and bYFV$_{ES504}$ complexed with either 5A Fab or 2C9 Fab with I3 symmetry applied. Maps are radially coloured according to the following: 0-150 Å red, 151-195 Å yellow, 196-210 Å green, 211-225 Å cyan, 226-240 Å blue Å, 241-255 Å orchid. **c** Cryo-EM density map and atomic model of bYFV$_{17D}$ complexed with 2C9 Fab. The density map was obtained via symmetry expanded ASU reconstruction and is shown in grey (E) and blue (variable region of 2C9 Fab), with the atomic model shown in red (E-DI), yellow (E-DII), blue (E-DIII) and grey (2C9 Fab). The N151 glycan is shown in green. **d** Top view of bYFV$_{17D}$:2C9 E dimer atomic model. **e** Comparison of the bYFV$_{17D}$:2C9 cryo-EM near-atomic resolution model and the YFV$_{17D}$ X-ray crystal structure (PDB:6IW4) in pink. Dotted lines are illustrating the E protein curvature differences between these two models.

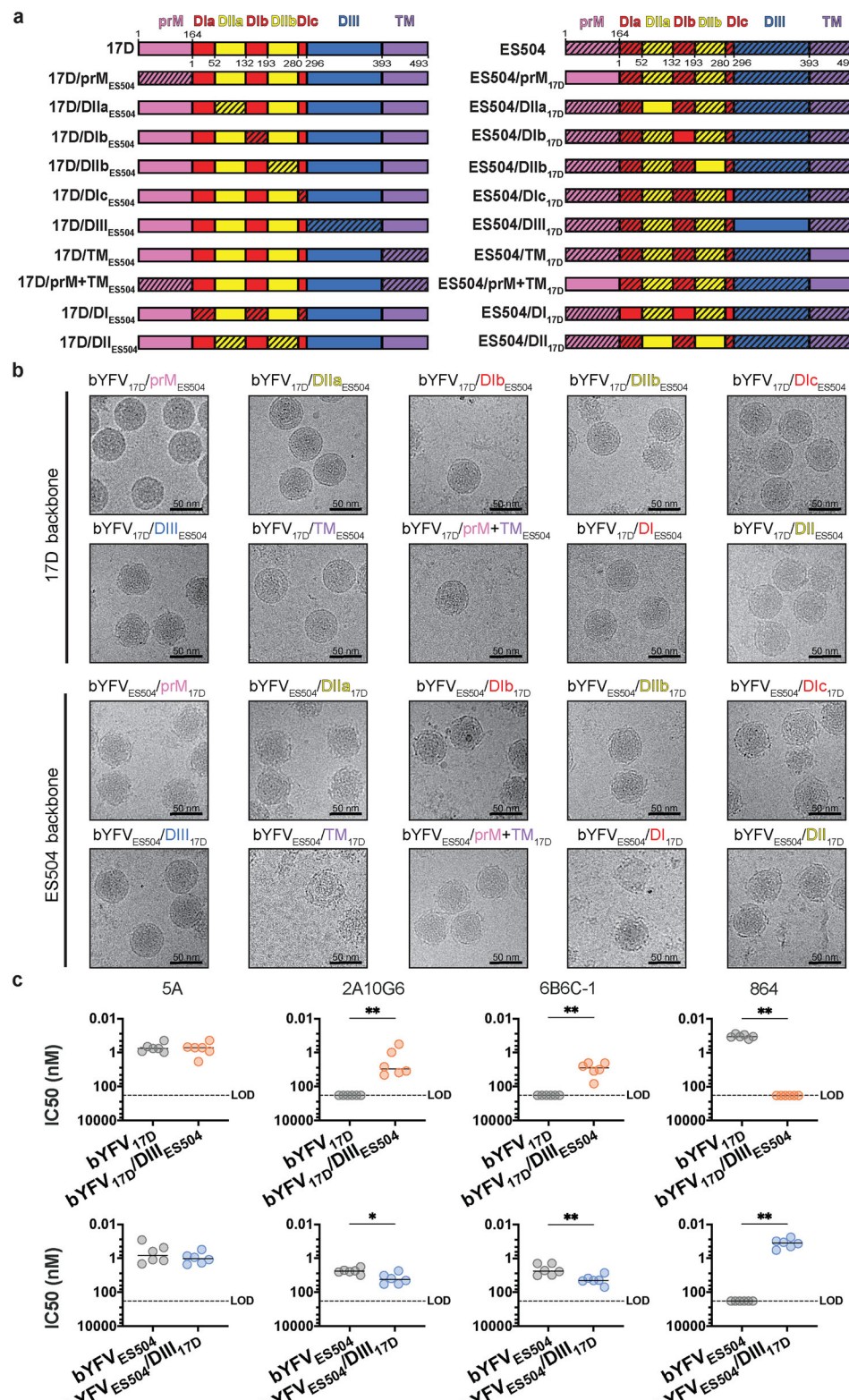

**Fig. 3 | Structural and antigenic characterisation of bYFV prME chimeras.**
**a** Schematic of the prM and E genes of the bYFV$_{17D/ES504}$ prME chimeras.
**b** Representative micrographs of purified bYFV$_{17D/ES504}$ prME chimera particles at 4 °C. **c** IC50 values of recombinant hIgG1 anti-YFV or anti-flavivirus mAbs against the bYFV DIII chimeras. Neutralisation was determined via FRNTs on C6/36 (*Aedes albopictus*) cells. Each symbol represents a technical replicate from three biological replicates (*n* = 6) and lines indicate group medians. Parental bYFV$_{17D}$ and bYFV$_{ES504}$

controls are from Fig. 1E. The bYFV DIII chimeras were compared with parental viruses by two-tailed Mann-Whitney tests on GraphPad Prism 9. **\*\****p* ≤ 0.002, **\****p* ≤ 0.03. For 2A10G6: *p* = 0.002 (17D vs. 17D/DIII$_{ES504}$) and *p* = 0.015 (ES504 vs. ES504/DIII$_{17D}$), 6B6C−1: *p* = 0.002 (17D vs. 17D/DIII$_{ES504}$) and *p* = 0.009 (ES504/DIII$_{17D}$), 864: *p* = 0.002 (17D vs. 17D/DIII$_{ES504}$) and *p* = 0.002 (ES504 vs. ES504/DIII$_{17D}$). Source data is provided as a Source Data file. *LOD* limit of detection.

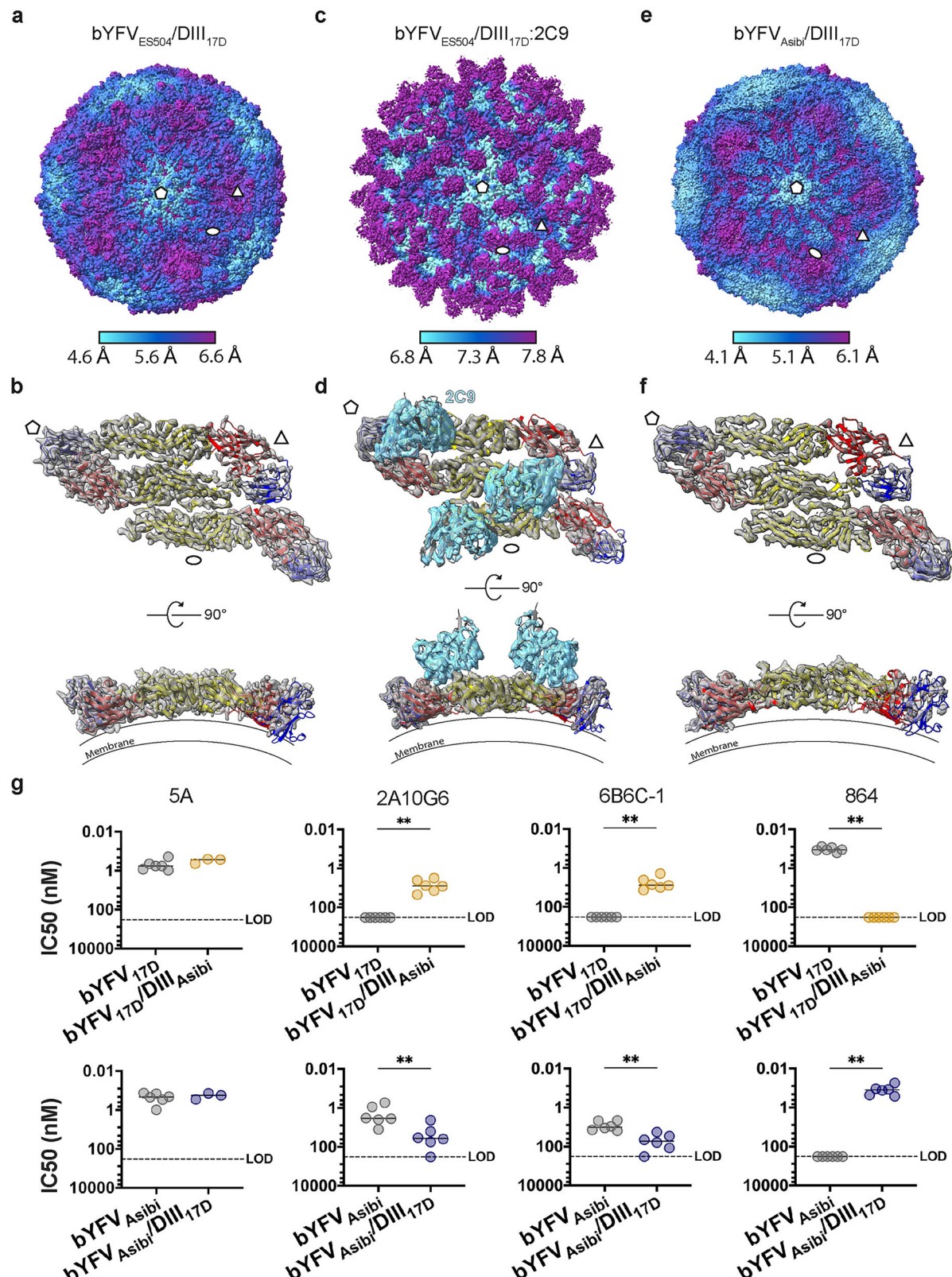

cryo-EM map was used to build an atomic model of the bYFV$_{ES504}$/DIII$_{17D}$ ASU, which differed from the bYFV$_{17D}$:2C9 ASU model at several expected sites (Fig. 4b, Supplementary Fig. 14).

To directly compare the structures of bYFV$_{17D}$ and bYFV$_{ES504}$/DIII$_{17D}$, we resolved the structure of bYFV$_{ES504}$/DIII$_{17D}$ complexed with Fab 2C9 to a resolution of 8.1 Å (Fig. 4c, Supplementary Figs. 15 and 16 and Supplementary Table 5). 2C9 Fab occupies all 180 sites on the

surface of bYFV$_{ES504}$/DIII$_{17D}$ (Fig. 4c). Next, we used the symmetry expanded local reconstruction method to resolve the bYFV$_{ES504}$/DIII$_{17D}$:2C9 ASU structure to 3.6 Å (Fig. 4d, Supplementary Fig. 16 and Supplementary Table 3). Similar to the bYFV$_{ES504}$/DIII$_{17D}$ ASU structure, the bYFV$_{ES504}$/DIII$_{17D}$:2C9 ASU map is not well defined at the icosahedral three-fold axis (Fig. 4d). An atomic model of the bYFV$_{ES504}$/DIII$_{17D}$:2C9 ASU was then built and compared to the bYFV$_{17D}$:2C9 ASU

**Fig. 4 | Cryo-EM of bYFV$_{ES504}$/DIII$_{17D}$ and bYFV$_{Asibi}$/DIII$_{17D}$. a** Cryo-EM density map of bYFV$_{ES504}$/DIII$_{17D}$ with I3 symmetry applied and coloured according to its local resolution. **b** Cryo-EM reconstruction and atomic model of the bYFV$_{ES504}$/DIII$_{17D}$ ASU. **c** Cryo-EM density map of bYFV$_{ES504}$/DIII$_{17D}$:2C9 with I3 symmetry applied and coloured according to its local resolution. **d** Cryo-EM reconstruction and atomic model of bYFV$_{ES504}$/DIII$_{17D}$:2C9 ASU. **e** Cryo-EM density map of bYFV$_{Asibi}$/DIII$_{17D}$ with I3 symmetry applied and coloured according to its local resolution. **f** Cryo-EM reconstruction and atomic model of bYFV$_{Asibi}$/DIII$_{17D}$ ASU. In (**e**) and (**f**) the pentagons, triangles, and ovals represent the icosahedral five-, three- and two-fold axes, respectively. In (**b**), (**d**), and (**f**) the density map was obtained via symmetry expanded ASU reconstruction using *cis*TEM2 and is shown in either grey

(E) or blue (2C9 Fab), with the atomic models shown in red (E-DI), yellow (E-DII), blue (E-DIII) and grey (2C9 Fab). **g** IC50 values of recombinant hIgG1 anti-YFV or anti-flavivirus mAbs against the bYFV DIII chimeras. Neutralisation was determined via FRNTs on C6/36 (*Aedes albopictus*) cells. Lines indicate group medians, and each symbol represents a technical replicate from three biological replicates. For bYFV$_{17D}$/DIII$_{Asibi}$ and bYFV$_{Asibi}$/DIII$_{17D}$ against 5A $n = 3$, and for all other groups $n = 6$. Parental bYFV$_{17D}$ and bYFV$_{Asibi}$ controls are from Fig. 1E. bYFV DIII chimeras were compared to parental viruses by two-tailed Mann-Whitney tests on GraphPad Prism 9. **$p \leq 0.002$. For 2A10G6: $p = 0.002$ (17D vs 17D/DIII$_{Asibi}$) and $p = 0.004$ (Asibi vs Asibi/DIII$_{17D}$), 6B6C−1 and 864: $p = 0.002$ (17D vs 17D/DIII$_{Asibi}$ and Asibi vs. Asibi/DIII$_{17D}$). LOD = limit of detection. Source data is provided as a Source Data file.

atomic model (Fig. 4d, Supplementary Fig. 14). Both models were very similar, with minor discrepancies in several of the surface loops as well as with the position of DIII (Cα RMSD: 1.25 Å).

**bYFV$_{Asibi}$/DIII$_{17D}$ recapitulates the findings of bYFV$_{ES504}$/DIII$_{17D}$**
Given that DIII of 17D was critical in improving the particle homogeneity of bYFV$_{ES504}$, we next examined whether swapping DIII between bYFV$_{17D}$ and bYFV$_{Asibi}$ would have a similar effect. As before, these DIII swap viruses were generated using CPER, gradient purified, and sequence confirmed. Both chimeric viruses were then imaged via cryo-EM. Remarkably, the bYFV$_{17D}$/DIII$_{Asibi}$ particles most closely resembled the phenotype of the virulent bYFVs, whilst the bYFV$_{Asibi}$/DIII$_{17D}$ particles most closely resembled the phenotype of bYFV$_{17D}$ (Supplementary Fig. 15). We then resolved the 3D structure of bYFV$_{Asibi}$/DIII$_{17D}$ to a resolution of 5.5 Å (Fig. 4e, Supplementary Figs. 17 and 18 and Supplementary Table 5). The symmetry expansion method was also used to further refine the ASU of bYFV$_{Asibi}$/DIII$_{17D}$ to 3.6 Å (Fig. 4f, Supplementary Fig. 18 and Supplementary Table 3). Similar to previous ASU maps, the bYFV$_{Asibi}$/DIII$_{17D}$ ASU map shows lower local resolution at the icosahedral three-fold axis, as well as no clear TM domains for M or E (Supplementary Fig. 18). This cryo-EM map was then used to build an atomic model of the bYFV$_{Asibi}$/DIII$_{17D}$ ASU, which was found to be highly similar to the bYFV$_{ES504}$/DIII$_{17D}$ ASU model with a Cα RMSD of 0.85 Å (Fig. 4f, Supplementary Figs. 14 and 18).

To further validate that DIII is contributing to the observed antigenic differences between vaccine and virulent strains of YFV, we assessed the neutralisation activity of 5A, 2A10G6, 6B6C-1 and 864 against bYFV$_{Asibi}$/DIII$_{17D}$ and bYFV$_{17D}$/DIII$_{Asibi}$. The exchange of DIII between bYFV$_{17D}$ and bYFV$_{Asibi}$ had the same observed effect on the neutralisation profile of 5A, 2A10G6, 6B6C-1 and 864, as the exchange of DIII between bYFV$_{17D}$ and bYFV$_{ES504}$ (Fig. 4g). This further validates that E-DIII greatly affects the antigenic landscape of YFV.

**R380 plays a crucial role in the structure and antigenicity of YFV**
Using single site mutagenesis, we then sought to identify if a specific residue within DIII was driving the observed phenotypes. Given that DIII of YFV$_{17D}$ stabilised bYFV$_{ES504}$ and bYFV$_{Asibi}$, we were able to focus on sites where both YFV$_{ES504}$ and YFV$_{Asibi}$ differed to YFV$_{17D}$. A comparison of the amino acid sequences of YFV$_{17D}$, YFV$_{ES504}$ and YFV$_{Asibi}$ revealed four amino acids in E-DIII where both YFV$_{ES504}$ and YFV$_{Asibi}$ differ to YFV$_{17D}$ (Fig. 5a). To assess if any of these four amino acids are solely responsible for the structural and antigenic variation observed thus far, we generated a panel of bYFV$_{17D}$ and bYFV$_{ES504}$ chimeras with amino acid substitutions at each of these four residues. These bYFV prME mutant chimeras were generated via CPER, gradient purified, sequence confirmed and then imaged using cryo-EM (Fig. 5b and Supplementary Fig. 19). The bYFV$_{17D}$ and bYFV$_{ES504}$ chimeras with amino acid substitutions at residues 299, 305 and 325 all resemble their parental phenotypes (Fig. 5b). However, bYFV$_{17D}$ R380T most closely resembles the heterogenous bumpy phenotype of bYFV$_{ES504}$ virions, and bYFV$_{ES504}$ T380R most closely resembles the smooth phenotype of bYFV$_{17D}$ virions (Fig. 5b).

Next, we assessed the effect of these single amino acids on the neutralisation profile of 5A, 2A10G6, 6B6C-1 and 864. The neutralisation activity of 5A was not affected by any of the amino acid substitutions, which was expected as they do not lie within 5As DII epitope (Fig. 5c)[26]. However, the neutralisation activity of both 2A10G6 and 6B6C-1 was significantly affected by the substitutions at amino acid 380. bYFV$_{17D}$ R380T is significantly more sensitive to neutralisation by both 2A10G6 and 6B6C-1, whilst bYFV$_{ES504}$ T380R is significantly less sensitive to neutralisation by both 2A10G6 and 6B6C-1, when compared with the other bYFV$_{ES504}$ mutant chimeras (Fig. 5c). The introduction of ES504 residues S305 and P325 in bYFV$_{17D}$ significantly reduced the neutralisation activity of mAb 864; accurately reflecting the known residues that are critical for 864 binding[44]. Of note, the introduction of 17D residues F305 and S325 in bYFV$_{ES504}$ did not restore any neutralisation activity by 864, suggesting that both residues may be required for mAb 864 binding (Fig. 5c).

The 3D structure of bYFV$_{ES504}$ T380R was resolved to a resolution of 5.7 Å (Fig. 6a, Supplementary Figs. 20 and 21 and Supplementary Table 5). The symmetry expanded local reconstruction method was used to resolve the ASU of bYFV$_{ES504}$ T380R to an initial resolution of 3.2 Å. To help elucidate all particle stabilising interactions that might be mediated by T380R, the symmetry expanded map was further refined after a focussed 3D classification around the previously ill-defined icosahedral three-fold axis. This resulted in two final maps with the three-fold axis E protein in two discrete states, which we designated up and down (Supplementary Fig. 21). While the global resolution of these maps are similar to the previous, the three-fold axis E protein is well resolved in the down map, allowing accurate modelling of the E protein in all three positions (Fig. 6b, Supplementary Fig. 21 and Supplementary Table 3). Residue R380, which is located on a non-surface exposed loop of E-DIII, makes contacts with neighbouring E proteins at three unique inter-raft interfaces (Fig. 6c, d). The five-fold axis E protein R380 residue makes two intermolecular bonds with the backbone carbonyl groups of D382 and R384 and the two-fold axis E protein R380 residue forms a salt bridge with E87 (Fig. 6d). In the down conformation, the three-fold axis E protein R380 residue makes a bond with the side chain hydroxyl of T133, while this contact is not observed in the up position. The atomic model of the bYFV$_{ES504}$ T380R ASU was then compared with the atomic model of the bYFV$_{ES504}$/DIII$_{17D}$ ASU and was found to be highly similar (Fig. 6e, rigid body whole ASU Cα RMSD: 0.82 Å).

**bYFV$_{Asibi}$ T380R recapitulates the findings of bYFV$_{ES504}$ T380R**
To further validate these findings, bYFV$_{Asibi}$ with a T380R mutation was generated via CPER, purified and imaged via cryo-EM (Supplementary Fig. 22). 3D reconstructions of the bYFV$_{Asibi}$ T380R, as well as the parental bYFV$_{Asibi}$, were resolved by SPA to resolutions of 8.6 Å and 14.5 Å, respectively (Supplementary Fig. 22 and Supplementary Table 5). The cryo-EM structure of bYFV$_{Asibi}$ T380R resembles that of bYFV$_{ES504}$ T380R and displays the typical flavivirus architecture of 180 E proteins arranged in a herringbone-like pattern on the virion surface. In contrast, the cryo-EM density map of bYFV$_{Asibi}$ lacks any typical

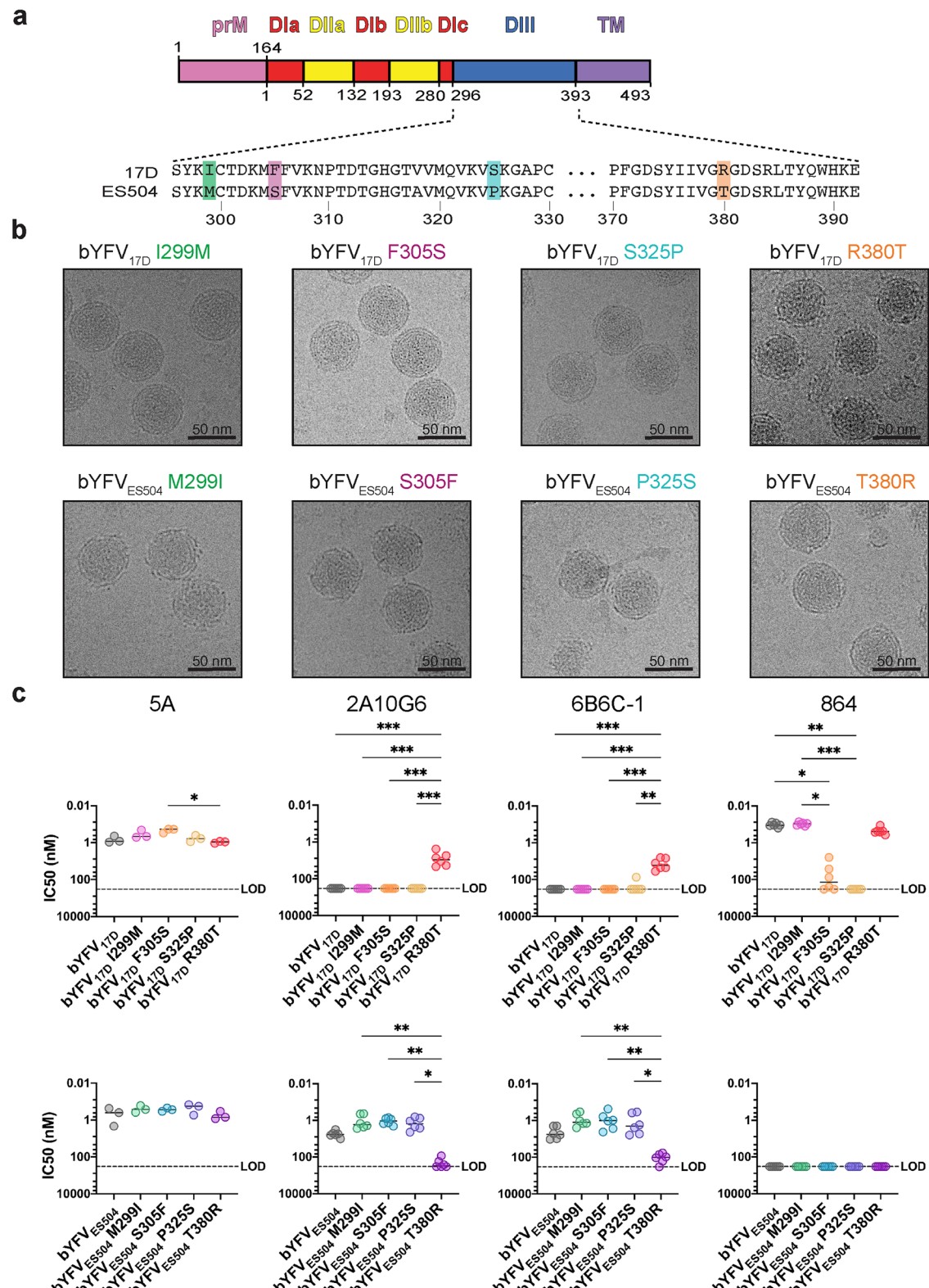

**Fig. 5 | Structural and antigenic characterisation of bYFV DIII mutant chimeras.**
**a** Schematic of the YFV genome illustrating the four amino acids in DIII where
YFV$_{17D}$ differs to both YFV$_{ES504}$ and YFV$_{Asibi}$. **b** Representative micrographs of the
bYFV DIII mutant chimeras at 4 °C. **c** IC$_{50}$ values of recombinant hIgG1 anti-YFV or
anti-flavivirus mAbs against the bYFV DIII mutant chimeras. Neutralisation was
determined using FRNTs on C6/36 (*Aedes albopictus*) cells. Each symbol represents
a technical replicate from three biological replicates (*n* = 6) and lines indicate group
medians. Parental bYFV$_{17D}$ and bYFV$_{ES504}$ controls are from Fig. 1E. Significance was
determined via Kruskal-Wallis tests with Dunn's multiple comparisons test on

GraphPad Prism 9.0. ***$p \leq 0.0002$, **$p \leq 0.002$, *$p \leq 0.03$. For 5A: $p = 0.035$ (17D
F305S vs. 17D R380T), 2A10G6: $p = 0.002$ (all 17D viruses vs. 17D R380T), $p = 0.004$
(ES504 M299I vs. ES504 T380R), $p = 0.016$ (ES504 S305F vs. ES504 T380R), $p = 0.01$
(ES504 P325S vs. ES504 T380R), 6B6C-1: $p = 0.0004$ (17D, 17D I299M and 17D
F305S vs 17D T380R), $p = 0.003$ (17D S325P vs 17D R380T), $p = 0.002$ (ES504 M299I
vs. ES504 T380R), $p = 0.005$ (ES504 S305F vs. ES504 T380R), $p = 0.040$ (ES504
P325S vs. ES504 T380R), 864: $p = 0.029$ (17D vs. 17D F305S), $p = 0.002$ (17D vs. 17D
S325P), $p = 0.01$ (17D I299M vs. 17D F305S), $p = 0.0005$ (17D I299M vs. 17D S325P).
Source data is provided as a Source Data file. LOD limit of detection.

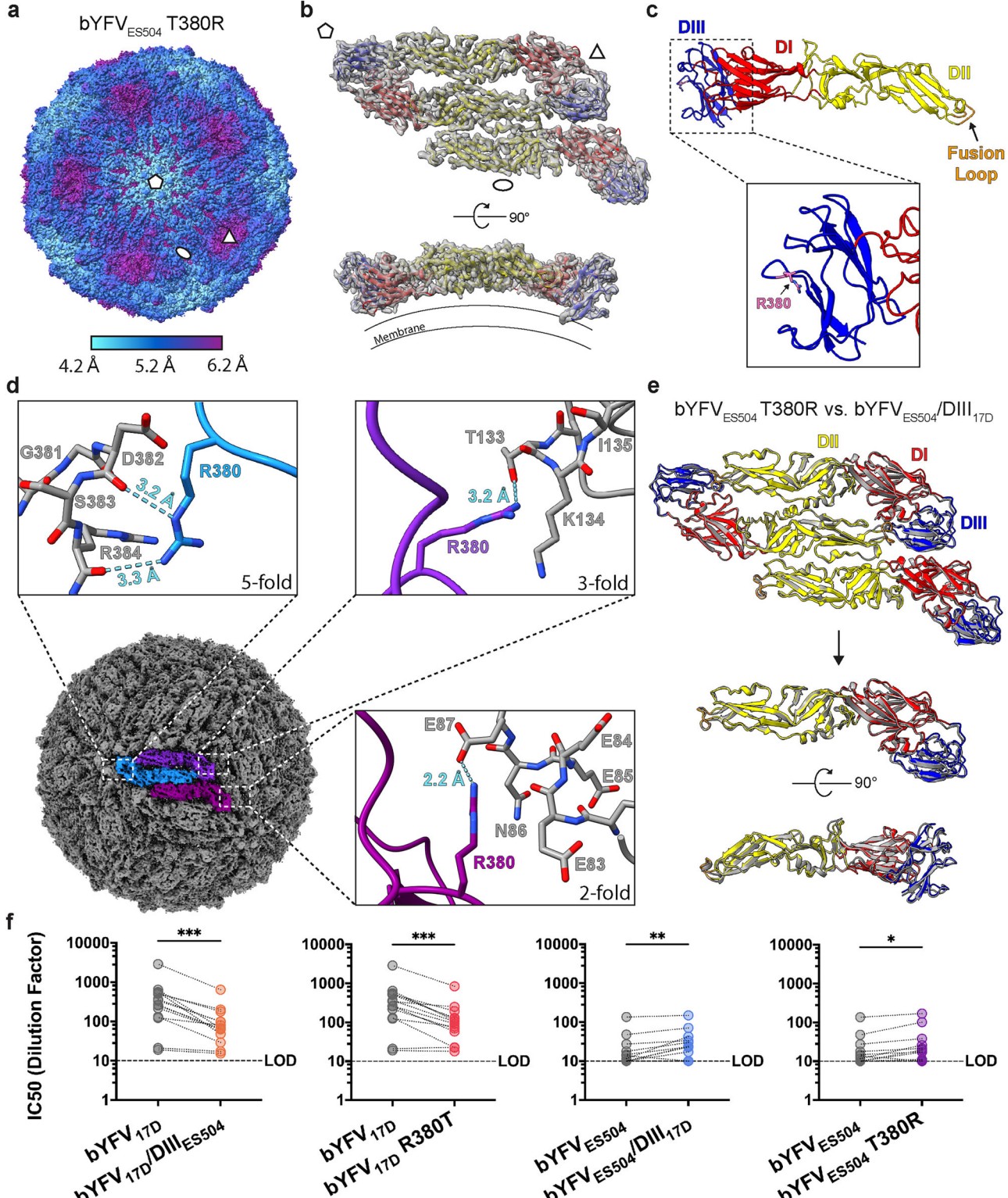

**Fig. 6 | Characterisation of bYFV$_{ES504}$ T380R. a** Cryo-EM reconstruction of bYFV$_{ES504}$ T380R with I3 symmetry applied and coloured according to its local resolution. **b** Cryo-EM reconstruction and atomic model of bYFV$_{ES504}$ T380R ASU. The density map was obtained by symmetry expanded ASU reconstruction using *cis*TEM2 and is shown in grey, with the atomic model shown in red (E-DI), yellow (E-DII) and blue (E-DIII). In (**a**) and (**b**) the pentagons, triangles, and ovals represent the icosahedral five-, three- and two-fold axes, respectively. **c** Atomic model of bYFV$_{ES504}$ T380R E monomer with R380 highlighted in dark pink and the fusion loop highlighted in orange. **d** Contacts between the five-fold (blue), three-fold (dark purple) and two-fold (light purple) axes E protein R380 residues and the adjacent E

proteins. Bonds are depicted by blue dotted lines. **e** Comparison of the bYFV$_{ES504}$ T380R and bYFV$_{ES504}$/DIII$_{17D}$ (in grey) ASU atomic models, aligned at the five-fold axis E protein. Below is a comparison of the three-fold axis E protein. **f** Serum neutralising titres of human YFV$_{17D}$ vaccinees ($n = 14$) against the bYFV DIII chimeras. Neutralisation was determined using FRNTs on C6/36 (*Aedes albopictus*) cells. Significance was determined using two-tailed Wilcoxon matched-pairs signed rank test on GraphPad Prism 9.0. ***$p \leq 0.0002$, **$p \leq 0.002$, *$p \leq 0.03$. $p = 0.0001$ (17D vs 17D/DIII$_{ES504}$), $p = 0.0004$ (17D vs. 17D R380T), $p = 0.009$ (ES504 vs. ES504/DIII$_{17D}$), $p = 0.017$ (ES504 vs. ES504 T380R). LOD = limit of detection. Source data is provided as a Source Data file. LOD limit of detection.

mature flavivirus characteristics (Supplementary Fig. 22). To further validate that residue 380 is contributing to the observed antigenic differences between vaccine and virulent strains of YFV, we assessed the neutralisation activity of 5A, 2A10G6 and 6B6C-1 against bYFV$_{Asibi}$ T380R (Supplementary Fig. 22). As observed with bYFV$_{ES504}$ T380R, bYFV$_{Asibi}$ T380R is significantly less sensitive to neutralisation by both 2A10G6 and 6B6C-1, when compared to the parental bYFV$_{Asibi}$ virus (Supplementary Fig. 22). Taken all together, these results suggest that residue 380 is playing an important role in not only the structure of YFV, but its antigenic landscape as well.

### R380 influences bYFV neutralisation by YFV$_{17D}$ vaccinee sera

Next, we assessed the effect of DIII and residue 380 on the neutralisation potential of YFV$_{17D}$ vaccinee sera. Comparison of the serum neutralising titres showed a significant decrease in neutralisation of both bYFV$_{17D}$/DIII$_{ES504}$ and bYFV$_{17D}$ R380T when compared with the parental bYFV$_{17D}$ (Fig. 6f). Conversely, there was a significant increase in neutralisation of both bYFV$_{ES504}$/DIII$_{17D}$ and bYFV$_{ES504}$ T380R when compared with the parental bYFV$_{ES504}$ (Fig. 6f).

## Discussion

Here, we utilised a chimeric virus platform to reveal high resolution structures of vaccine and virulent envelope proteins of YFV. Despite obtaining pure and mature preparations of these chimeric virions, stark differences in particle morphology and homogeneity were observed by cryo-EM. Cryo-EM reconstructions, both with and without icosahedral symmetry imposed, revealed a classical herringbone E raft assembly for the bYFV$_{17D}$ virions. In contrast, the surface features of bYFV$_{ES504}$ virions could not be resolved well and were reminiscent of DENV virions after incubation at high temperatures[45,46]. However, the virulent YFV virions were grown at 28 °C, and subsequently purified and stored at 4 °C, prior to cryo-EM imaging. This would suggest, that unlike DENV virions[45], bumpy heterogenous virions appear to be an intrinsic feature of pathogenic YFV in both mosquito and human hosts.

Mimicking the differences in neutralisation sensitivity observed between smooth and bumpy DENV, we observed a correlation between the virion structures of vaccine and virulent strains of YFV and their sensitivity to neutralisation by mAbs that target cryptic epitopes. We found that bYFV$_{ES504}$ and bYFV$_{Asibi}$ are sensitive to neutralisation by the cryptic FLE mAbs 2A10G6 and 6B6C-1, while bYFV$_{17D}$ is resistant to neutralisation by these mAbs. There is a growing consensus in the literature that cryptic E protein epitopes are much more accessible on the surface of bumpy virions in comparison with smooth and compact virions[47–51]. In agreement with this hypothesis, we were able to visualise efficient binding of 2A10G6 Fabs to the bumpy bYFV$_{ES504}$ virions, but only a few Fab like densities were associated with the smooth bYFV$_{17D}$ virions, consistent with an occupancy level below that required for neutralisation. In contrast, FLE mAbs bound to purified mature preparations of all three bYFV strains, confirming that FLE mAb neutralisation is not influenced by local sequence variability. Instead this is likely due to destabilisation of the purified virions once surface-immobilised and probed in the presence of a gentle surfactant in ELISA. These results highlight the importance of performing functional assays to examine epitope availability within the context of the intact infectious virion as has been previously observed for other flaviviruses[52].

Exhaustive domain swap interrogation between bYFV$_{17D}$ and bYFV$_{ES504}$ revealed that DIII is a key determinant of virion icosahedral assembly and FLE mAb neutralisation. Exchange with DIII of YFV$_{17D}$ allowed for high resolution elucidation of the bYFV$_{ES504}$ virion structure in both unbound and 2C9 (E-DII mAb) bound forms, demonstrating high similarity to bYFV$_{17D}$. As DIII makes direct contact with the FL of the opposing E monomer, it may influence dimer stability and thus offers a potential mechanism for DIII-mediated modulation of FL exposure. However, the FL-proximal AB loop of DIII (residues 308-321,

Supplementary Fig. 10) is largely conserved between YFV$_{17D}$ and YFV$_{ES504}$; therefore, the particle stabilisation induced by swapping DIII is more likely due to inter-dimer or inter-raft interactions. Furthermore, incorporation of YFV$_{17D}$ DIII into bYFV$_{Asibi}$ also switched virion homogeneity and FLE mAb neutralisation sensitivity phenotypes. This shared stabilisation of divergent strains allowed us to focus on four single DIII residues that are shared amongst both virulent strains of YFV but differ in YFV$_{17D}$: I299M, F305S, S305P and R380T. Of these four substitutions, only R380T was observed to induce sensitivity of the bYFV$_{17D}$ virions to FLE mAb neutralisation. Mirroring this result, the single change T380R in the bYFV$_{ES504}$ and bYFV$_{Asibi}$ virions reduced the neutralisation potency of the FLE mAbs. Introduction of R380 into bYFV$_{ES504}$ also resulted in more homogenous virions, allowing for its ASU to be resolved to a high resolution. In this context R380 makes contacts at three unique inter-raft interfaces: at the icosahedral five- and three-fold symmetry axes, as well as centrally between the five- and three-fold axes. The five-fold axis E protein R380 makes three hydrogen bonds with the Cα backbone of an adjacent five-fold axis DIII and the two-fold axis E protein R380 residue forms a salt bridge with E87 from an adjacent DII. Interestingly, two discrete positions of the three-fold axis E protein were revealed using a focussed 3D classification of the symmetry expanded ASU, one with the three-fold axis E protein in an up conformation and one with the three-fold E protein in a down conformation. The three-fold axis R380 residue is only able to make a contact with the adjacent E protein whilst it is in the down conformation, which is mediated through a hydrogen bond with the conserved T133 residue from an adjacent E-DI. Given the binary-like flip in sensitivity to FLE mAb neutralisation for the residue 380 mutant viruses, it is likely that the up and down conformations of the three-fold axis E protein reflect transitory states that are stabilised by the other, relatively stronger, inter-raft interactions at the other symmetry axes.

Our findings have broad implications for flavivirus vaccine design, where FLE antibodies are thought to be a major driver of antibody dependent enhancement (ADE) of infection[53,54]. Given that the introduction of R380 reduces FLE exposure, this will allow the development of next-generation YFV vaccines that use contemporary strain sequences while retaining lower potential for ADE. This approach contrasts from other designs to reduce FLE exposure in DENV and ZIKV, which have focused on dimer stabilisation[55,56]. Instead, we propose that the three inter-raft interactions mediated by R380 act together to lock the smooth mature virion structure, and that this conformational stabilisation propagates down to the dimer level. This provides a new framework to design flavivirus antigens at the particle level, reducing FLE exposure and increasing the presentation of quaternary epitopes, which are the targets for potent neutralising mAbs for other flaviviruses[30–32,57–60].

While quaternary epitope specific antibodies have yet to be characterised for YFV, our findings highlight the need to also investigate the FLE response in YFV infection and vaccination. Notably, all R380 interactions are with highly conserved residues present in all known YFV genotypes, offering a universal strategy to stabilise diverse YFV viruses for structural and antigenic characterisation, which will greatly assist future structural determination of quaternary epitopes. In agreement with the generalisable nature of this finding, we show that Asibi derived mature particles can also be stabilised with the introduction of R380 (Supplementary Fig. 22). Interestingly, although we observed mirrored reciprocity of the neutralisation sensitivity between vaccine and virulent strains when swapping residue 380, the magnitude of the effect was different, suggesting that other residues or combinations also play a role in FLE exposure. Of note, we assessed a panel of human YFV$_{17D}$ vaccine sera and observed a significant drop in neutralisation for both bYFV$_{ES504}$ and bYFV$_{Asibi}$, compared with bYFV$_{17D}$. Similar results have recently been reported by Haslwanter and colleagues[20]. However, whilst their work focussed on a glycosylation

site unique to the South American strains of YFV, our work demonstrates that the quaternary virion structure also plays a significant role in resistance to vaccine induced antibodies. This effect is broader than that attributed to the genotype-specific glycan, and offers an explanation for the lower potency of vaccine induced neutralisation against YFV$_{Asibi}$ compared to YFV$_{17D}$, which was observed both here and in the Haslwanter et al. study. Our findings also suggest that YFV$_{17D}$ induces E dimer and quaternary epitope specific antibodies that are vaccine specific. Recent work supports this hypothesis, with a dimeric YFV E construct able to significantly remove most YFV$_{17D}$ neutralising antibodies compared to monomeric E[61]. Future mAb discovery focused on convalescent YFV survivors is needed, and our findings show that such work should include whole virion based studies to fully capture the striking quaternary-level differences that drive neutralisation sensitivity between vaccine and virulent strains of YFV.

We conclude that the surface architecture of mature virions composed of E proteins from pathogenic YFV strains, both South American and African genotypes, is intrinsically heterogenous and that this has implications for antibody recognition and neutralisation. The FLE is a dominant flavivirus epitope, is highly conserved across the diverse flavivirus genus, and is strongly correlated with ADE in DENV infection. Understanding the potential for FLE responses to provide protection or enhancement for virulent YFV is urgently needed to inform both YFV vaccine redesign, as well as understand interactions with co-circulating flaviviruses and current approved vaccines. While we and others show that the 17D vaccine is an imperfect match for the current circulating strains of YFV, our work also shows that the serial passaging used to generate 17D resulted in a highly stable particle with low FLE exposure. Therefore, it is imperative that work to design a more optimal YFV vaccine takes into account the potential for FLE antibody generation, as well as cross-reactivity to other important flaviviruses both current and emerging.

## Methods

### Ethics statement
The research was approved by the University of Queensland Human Ethics Unit (2021/HE000021 – Using Human B cells to Produce Monoclonal Antibodies). All participants in this study were volunteers, provided written informed consent and approved the publication of indirect identifiers. Human volunteers self-reported sex and age.

### Cell culture
C6/36 (*Aedes albopictus*, ATCC – CL1660) cells were maintained at 28 °C in RPMI media containing 25 mM HEPES and L-glutamine (Gibco), and supplemented with 5% heat-inactivated foetal bovine serum (FBS, Bovogen), 100 U/mL penicillin and 100 μg/mL streptomycin (Gibco). ExpiCHO cells (Gibco) were maintained in ExpiCHO expression medium (Gibco) at 37 °C with 7.5% $CO_2$.

### Human samples
Blood samples were collected from healthy adults (8 males and 6 females) with a history of YFV vaccination (details in Supplementary Table 2). Whole blood was collected in BD Vacutainer SST$^{TM}$ tubes (BD). For serum separation, samples were incubated for 1 h at room temperature and then centrifuged at 4000 × $g$ for 14 min. After centrifugation, serum was aliquoted, heat-inactivated at 56 °C and stored at -80 °C until use.

### Generation of chimeric viruses
Chimeric viruses were generated using a circular polymerase extension reaction (CPER) methodology[34,35]. RNA of BinJV (GenBank: MG587038.1) and YFV$_{17D}$ (GenBank: KF769015.1) was extracted from infected cell supernatant using NucleoSpin RNA Columns (Machery-Nagel) and converted to complementary DNA using SuperScript III Reverse Transcriptase (Invitrogen). Overlapping double stranded DNA

(dsDNA) fragments covering the capsid and NS1-5 genes, as well as the untranslated regions (UTRs) were amplified from the BinJV cDNA. An overlapping dsDNA fragment covering the prM and E genes was amplified from the YFV$_{17D}$ cDNA. The prME fragments of YFV$_{ES504}$ (GenBank: KY885000.2) and bYFV$_{Asibi}$ (Genbank: AY640589.1) were generated by site-directed mutagenesis of the YFV$_{17D}$ cDNA. The prME fragments of the domain swap mutants were generated via overlapping PCR, and the prME fragments of the single-site mutants were generated via site-directed mutagenesis. Primers used to generate all fragments are provided in a separate Supplementary Data 1 file. The BinJV backbone fragments, YFV prME fragment and a linker fragment connecting the 5' and 3' UTRs were purified and added in equimolar proportions (0.1 pmol) to a PrimeSTAR GXL reaction and incubated under the following conditions: 98 °C for 30 secs (1 cycle), 98 °C for 10 secs, 55 °C for 20 secs, 68 °C for 13 mins (12 cycles) and 68 °C for 13 mins (1 cycle). The reaction was transfected into C6/36 cells using Effectene (Qiagen) or TransIT-LT (Mirus), as per the manufacturer's guidelines. Chimeric virus particles were recovered from the supernatant 10-14 days post transfection. Titres were determined via immunoplaque assays and prME genes were sequence confirmed using Sanger sequencing.

### Virus stocks
Virus stocks were generated by infecting C6/36 monolayers at a multiplicity of infection (MOI) of 0.01 in serum free RMPI 1640 media (Gibco). Infected C6/36 cells were incubated for 1.5 h at 28 °C and then supplemented with RPMI media containing FBS to a final concentration of 2%. Cells were incubated for a further 6-8 days at 28 °C. When a cytopathic effect was observed, cell culture supernatant was collected and clarified via centrifugation at 3000 × $g$ for 20 min at 4 °C. Virus stocks were stored at −80 °C until use.

### Recombinant antibody expression and purification
The variable region of the heavy chain and light chain of 5A[26], 2C9[36,42], 2A10G6[37,62], 6B6C-1[40] and 864[39,44] were in-fusion cloned into previously described hIgG1 mammalian expression vectors[63]. To generate 5A and 2C9 Fabs, a human rhinovirus 3C (HRV3C) protease site was introduced at the HC hinge region. 25 μg total of the HC and LC vectors were co-transfected into freshly passaged ExpiCHO cells at a density of 6×10$^6$ cells/mL, as per the manufacturer's instructions. At 7 days post-transfection, the ExpiCHO supernatant was harvested via centrifugation at 4800 × $g$ for 10 min and filter-sterilised using a 0.22 μm filter. Antibodies were purified using AKTA-FPLC affinity chromatography with a Protein A HP column (GE Healthcare). The column was washed with 25 mM Tris, 25 mM NaCl, pH 7.4 and eluted with 100 mM sodium citrate, 150 mM NaCl, pH 3.0. Fractions were immediately pH-adjusted with 1.5 M Tris-HCl pH 8.8. Recombinant mAbs were filter-sterilised using a 0.22 μm filter and buffer exchanged into phosphate buffered solution (PBS) using centrifugal filter units with a 30 kDa molecular weight cut off (Merck). To generate 5A and 2C9 Fabs, mAb containing the HRV3C site was digested for 48 h with in-house generated HRV3C protease at 4 °C. Fab was separated from undigested mAb and the Fc fragment by using a combination of protein A resin (IBA Lifesciences) and glutathione resin (Genscript). Flow through was collected, 0.22 μM filter-sterilised and buffer exchanged into PBS pH 7.4. Concentrations were determined using a Nanodrop 1000 spectrophotometer.

### Virus purification
C6/36 cells were infected with virus at a MOI of 0.01 in serum-free RPMI media, incubated at 28 °C for 1.5 h and then supplemented with RPMI media containing FBS to a final concentration of 2%. The supernatant was collected and replenished 3 times between days 4 and 13 post-infection, clarified via centrifugation (30 mins at 3000 × $g$), filtered through a 0.22 μm filter and stored at 4 °C until purification. The virus was precipitated by adding polyethylene glycol 8000 (final

concentration 8% *w/v*, Fisher Bioreagents) in NTE buffer (120 nM NaCl, 10 mM Tris-Base and 1 mM EDTA, pH 8) and incubated overnight at 4 °C before centrifugation at 12,000 × *g* for 1 h at 4 °C using an Avanti J-26 XPI JLA-10.500 rotor. The virus pellet was resuspended in NTE buffer and purified via a 20% *w/v* sucrose cushion centrifuged at 133,907 × *g* for 2 h using a Beckman Coulter Optima L-100 XP SW32Ti rotor. The virus containing pellet was resuspended in NTE buffer and incubated overnight before centrifugation at 10,000 × *g* at 4 °C for 10 mins. The resulting supernatant was further purified via a 25–40% *w/v* potassium tartrate gradient, centrifuged at 336,840 × *g* for 1 h at 4 °C using a Beckman Coulter Optima L-100 XP SW60Ti rotor. Virus bands were harvested, buffer exchanged into NTE via ultrafiltration in Amicon™ centrifugal filter units, and stored at 4 °C until use. Purity was determined by reduced SDS-PAGE followed by Coommassie Blue stain (ThermoFisher Scientific).

### Cryo-electron microscopy
An aliquot of 2 μL of purified virus at a concentration of 3 mg/mL was applied to glow discharged R3.5/1 or R1.2/1.3 holey carbon grids (Quantifoil). Vitrification conditions were optimised and the same conditions were used for each sample, using an EM GP2 (Leica Microsystems) set at 4 °C and 95% relative humidity, grids were pre-blotted for 10 secs, followed by blotting for 3 secs and immediately plunge-frozen in liquid ethane. Grids were transferred under liquid nitrogen to a CRYO ARM™ 300 (JEOL JEM-Z300FSC) transmission electron microscope operated at 300 kV, equipped with a K3 direct electron detector (Gatan) and an omega in-column energy filter operated at 20 eV (JEOL). Movies were recorded using SerialEM at 60,000x magnification[64]. The data collection parameters varied for each sample and are summarised in the Supplemental Information.

### Image processing and single-particle analysis
For datasets processed using RELION 3.1.3, the movies were binned two times and motion corrected using MotionCor2 (v. 1.1.0) and the CTF parameters of each image were estimated using CTFFIND (v. 4.1)[65–67]. Particles were automatically picked from all micrographs using a template generated from the 2D-classification of a small subset of manually picked particles. Following particle extraction, disordered particles were removed using reference-free 2D-classifications. A reconstruction of mature BinJV-Rocio_prME (generated in-house) was low-passed filtered to 60 Å and used as a reference for 3D-classification with I3 symmetry. The most ordered classes from the 3D-classifications were further refined using the RELION 3D auto-refine procedure. For majority of the reconstructions, a spherical mask excluding the solvent was applied to the half-maps. The FSC was calculated using the post-processing procedure in RELION or the Electron Microscopy Databank (EMDB) FSC server. FSC plots are shown in the Supplemental Information.

For symmetry expanded asymmetric localised reconstructions, the beam-induced motions were corrected and the contrast transfer function (CTF) parameters were estimated using *cis*TEM2[68]. Virus particles were automatically picked, extracted and subjected to 2D classifications. The initial 3D reconstructions of the icosahedral virion were carried out ab initio, followed by 3D refinement using *cis*TEM2. High-resolution asymmetric focused reconstructions were carried out by symmetry expansion using *cis*TEM2[41]. Each image was subtracted 60 times using each of the 60 icosahedral ASUs generating a dataset of isolated asymmetric units which were automatically centred. This dataset of asymmetric units was further refined in *cis*TEM2 without applying symmetry. Symmetry expansion, signal subtraction, and ASU particle image cropping were carried out using the program symmetry_expand_stack_and_par included with *cis*TEM2[41]. The ASU maps were sharpened using DeepEMhancer (bYFV_ES504 T380R ASU maps) or *cis*TEM2 (remaining ASU maps), and the FSC was calculated on *cis*TEM2 using only loose spherical masks (FSC plots shown in the Supplemental

Information)[69]. All cryo-EM figures were generated using ChimeraX (v. 1.2.5)[70].

### Fab crystallisation
Fab 2C9 was purified by size exclusion chromatography using an Enrich SEC650 10×300 column (BioRad) and concentrated to 10 mg/mL. Robotic screening in sitting drops of Fab 2C9 was conducted at the Monash Molecular Crystallisation Facility. Needle-like crystals were observed in three conditions: condition A: 0.1 M MES pH 6.0, 20% *w/v* PEG 6,000 and 0.01 M ZnCl₂; condition B: 12% *w/v* PEG 3,350, 0.1 M HEPES pH 7.5 and 0.005 M of CoCl₂, CdCl₂, MgCl₂ and NiCl₂; condition C: 0.1 M NaCl, 1.6 M (NH₄)₂SO₄ and 0.1 M HEPES pH 7.5. Optimisation of Condition C using streak seeding produced the best crystals in 1.3 M (NH₄)₂SO₄, 0.1 M HEPES pH 7.0, 0.1 M NaCl that diffracted to a resolution of 2.66 Å at the MX2 beamline at the Australian Synchrotron. Data was processed using XDS software package followed by Aimless and Pointless in the CCP4 software suite (v. 7.1.015)[71]. The structure was solved by molecular replacement using Phaser_MR (v. 2.8.3)[72] with the previously published structure of a Fab of 5.6 monoclonal mouse IgG1 (PDB: 7JVD) (85) as an initial model. Three molecules are present in the asymmetric unit. Rebuilding and refinement were conducted in Coot (v 0.8.9.2)[73] and PHENIX.refine (v. 1.19.2-4258)[74,75], respectively.

### Model building
The initial atomic model of bYFV_ES504 T380R was generated using ModelAngelo and rebuilt where needed[76]. For the remaining atomic models, the final model of bYFV_ES504 T380R was used as an initial model. In all models, the five-fold axis E monomer was built first and used as an initial model for the three- and two-fold axis E monomers, which were rebuilt as needed to account for the heterogeneity within the ASU. As the bYFV_ES504/DIII_17D, bYFV_ES504/DIII_17D:2C9 and bYFV_Asibi/DIII_17D ASU density maps were not well resolved at the three-fold symmetry axis, the five-fold DI and DIII were used instead. The initial atomic model of Fab 2C9 was generated using Alphafold2 and rebuilt where needed[77,78]. All modelling was performed using iterative cycles of manual building in COOT (v. 0.8.9.2) and ISOLDE (v. 1.2.1) and real-space refinements in PHENIX (v. 1.19.2)[73–75,79]. The geometry and quality of the models were evaluated through a combination of PHENIX and the wwPDB Validation System.

### Enzyme-linked immunosorbent assay (ELISA)
Nunc MaxiSorp 96-well plates were coated with purified virus particles at a concentration of 2 μg/mL (diluted in PBS) and incubated overnight at 4 °C. Plates were blocked with 150 μL of blocking buffer (milk diluent concentrate (KPL) in PBS with 1% Tween20) and incubated at room temperature for 1 h. Following incubation, the blocking buffer was removed, and five-fold serial dilutions of the primary antibodies were added to the wells (50 μL/well) and incubated for 1 h at 37 °C with 5% CO₂. The primary antibodies were removed, and the plates were washed 3 times in PBS with 1% Tween20. Goat anti-human horseradish peroxidase (HRP) conjugated antibody (Invitrogen, Cat #A18829, Lot #95-14-052722) at 0.2 μg/mL was added (50 μL/well), incubated for 1 h at 37 °C with 5% CO₂ and washed three times in PBS with 1% Tween20. Prewarmed 3,3′,5,5;-tetramethylbenzidine (TMB) solution (Life Technologies) was added to the plates (50 μL/well) and incubated until sufficient colour had developed (~3-5 mins). The reaction was stopped by the addition of 1 M sulphuric acid (25 μL/well) and absorbance read at 450 nm. Absorbance values were plotted using GraphPad Prism 9 (v. 9.1.0) software and the apparent affinity (K_d value) for each antibody was obtained using a one-site specific binding model.

### Focus reduction neutralisation tests
C6/36 cells were cultured in 96-well micro-culture plates (Nunc) at a density of $1 \times 10^7$ cells/well overnight at 28 °C with 5% CO₂. Five-fold serial dilutions of mAbs (starting at 100 μg/mL) were added to an equal

volume of virus (500 plaque-forming units (PFU)/mL). For FRNTs using human sera samples, three-fold serial dilutions of sera (starting at a 1:10 dilution) was added to an equal volume of virus. The mAb-virus/sera-virus mixture was incubated for 1 h at 37 °C with 7.5% $CO_2$, transferred onto cells (50 μL/well) and incubated for a further 2 h at 28 °C with 5% $CO_2$. 130 μL of overlay media was added to the cells and incubated for a further 48 to 72 h at 37 °C with 5% $CO_2$. Following incubation, the plates were fixed with ice-cold 80% acetone in PBS for 30–60 min at -20 °C. After the incubation, the acetone was removed, and the plates were dried. The plates were stained using mouse 4G4 (anti-flavivirus NS1) at 1 μg/mL as the primary antibody and anti-mouse IR800 (LI-COR Biosciences, Cat #926-32210, Lot #D40312-15) at 0.2 μg/mL as secondary antibody. Plates were scanned using the Odyssey CLx Infrared Imaging System (LI-COR Biosciences) at a resolution of 42 μm in the 800 nm channel. Foci were counted by eye and a 50% inhibitory concentration (IC50) was determined using a three-parameter [inhibitor] vs. response model in GraphPad Prism 9 (v. 9.1.0).

### Statistics and reproducibility

All statistical comparisons are indicated in the Figure legends. Curve-fitting and statistical analysis was done in GraphPad Prism 9 (v. 9.1.0). The representative micrographs of $bYFV_{17D}$, $bYFV_{ES504}$, $bYFV_{Asibi}$ $bYFV_{17D}$:5A, $bYFV_{17D}$:2C9, $bYFV_{ES504}$:5A and $bYFV_{ES504}$:2C9 are indicative of two independent biological replicates. The representative micrographs of the bYFV mutant chimeras are from one independent biological replicate.

### Reporting summary

Further information on research design is available in the Nature Portfolio Reporting Summary linked to this article.

## Data availability

The cryo-EM/X-ray crystallography maps and models generated in this study have been deposited in the Electron Microscopy Data Bank and the Protein Data Bank, respectively, under accession codes: $bYFV_{17D}$ (EMD-44278), $bYFV_{ES504}$ (EMD-44279), $bYFV_{17D}$:2C9 (EMD-44280), $bYFV_{ES504}$:2C9 (EMD-44281), $bYFV_{17D}$:5A (EMD-44282), $bYFV_{ES504}$:5A (EMD-44283), $bYFV_{ES504}$/$DIII_{17D}$ (EMD-44284), $bYFV_{ES504}$/$DIII_{17D}$:2C9 (EMD-44285), $bYFV_{Asibi}$/$DIII_{17D}$ (EMD-44286), $bYFV_{ES504}$ T380R (EMD-44287), $bYFV_{Asibi}$ (EMD-49804), $bYFV_{Asibi}$ T380R (EMD-49805), $bYFV_{17D}$:2C9 ASU (EMD-44288, PDB: 9B6U), $bYFV_{ES504}$/$DIII_{17D}$ ASU (EMD-44289, PDB: 9B6V), $bYFV_{ES504}$/$DIII_{17D}$:2C9 (EMD-44290, PDB: 9B6W), $bYFV_{Asibi}$/$DIII_{17D}$ ASU (EMD-44291, PDB: 9B6X), $bYFV_{ES504}$ T380R ASU (EMD-44292, PDB: 9B6Y) and 2C9 Fab (PDB: 9B8G). Other structures used in this study are available in the Protein Database under accession codes 6IW4 (X-ray crystal structure of $YFV_{17D}$) and 7JVD (Fab of 5.6 mAb). Source data are provided with this paper.

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

## Acknowledgements

This work was supported by NHMRC Project Grant APP1164216 and NHMRC Ideas Grant APP2004582. D.W was supported by CSL Centenary Fellowships. N.M was supported by UQ ARC DECRA. The authors acknowledge Drs. Matthias Flotenmeyer and Lou Brillault from the Centre for Microscopy and Microanalysis, University of Queensland (UQ) and the facilities. Data processing was supported by the High Performance Computing facility (HPC) provided by the Research Computing Centre, UQ. We thank Ian Mortimer (ITS Infrastructure Operations, UQ). Diffraction experiments were undertaken on the MX2 beamline at the Australian Synchrotron, part of ANSTO. We thank all blood donors.

## Author contributions

S.B., P.R.Y., R.A.H., J.H.-P., N.M. and D.W. conceived and designed the study. S.B., A.A.A. and N.N., generated and purified the virus samples. Y.S.L. and N.M. collected cryo-EM data. S.B., J.J., T.G., N.M. and D.W. processed cryo-EM data. S.B. and N.M performed cryo-EM model building. J.B., Y.T.T. and F.C. collected X-ray crystallography data and performed X-ray crystallography model building. S.B., C.A.P.S., B.L. and M.E.F., generated and purified antibodies. S.B. performed binding and neutralisation assays. S.B., J.J. and D.W. wrote the manuscript. All authors reviewed and contributed to the final manuscript.

## Competing interests

N.D.N., R.A.H., J.H.-P. and D.W. are inventors on patent application WO/2018/176075, relating to the chimeric BinJV platform. The remaining authors declare no competing interests.
