## [Transparent Peer Review file · Nature Communications]

A single residue in the yellow fever virus envelope protein modulates virion architecture and antigenicity

Corresponding Author: Dr Daniel Watterson

Version 0:

Reviewer comments:

Reviewer #1

(Remarks to the Author)

Bibby and colleagues report in this paper the cryoEM structure of chimeric yellow fever/Binjari viruses comparing yellow fever 17D vaccine virus (YFV-17D) with yellow fever wild type viruses (YFV-ES504 and YFV-Asibi). The authors describe a different conformation of the surface lattice of the vaccine and wild type chimera. They report that the vaccine and wild type chimera show different reactivity to well characterized antibodies suggesting different antigenicity that can be traced back to a specific mutation in DIII of the envelope protein.

Yellow fever virus is a highly pathogenic flavivirus whose spreading is controlled by vaccination with the very efficient live attenuated vaccine 17D. However recent outbreaks in Africa and South America have raised concern about the re-emergence of this virus and the ability of the vaccine to control it.

Thus, structural studies aiming to understand the antigen landscape of this virus are needed and necessary.

Nevertheless, while the virus studied in this work is relevant, the authors put a too strong emphasis on the “whole virion structure of yellow fever virus” (lines 90-91) which obscures the fact that in reality the reported structures are chimeras that recapitulate the surface lattice of yellow fever virus. This overstatement is pursued through the entire paper.

Several points are either overemphasized or not clear.

In particular:

- 1) The title seems too strong and shall include the mention of the chimera system.
- 2) The introduction would benefit from the presentation of the different yellow fever strains studied (17D, Asibi and ES504) and a workflow on how the Binjary system works.
- 3) The structural differences claimed in Fig.1 are not clearly visible from the micrographs or the 2D classes and the cryoEM reconstructions are not comparable in terms of resolution to appreciate a difference.
- 4) There is no mention of why the bYFVAsibi structure was not pursued/solved.
- 5) Lines 166-167: the density maps of bYFVES504 in complex with 5A or 2C9 antibody are described as incomplete and with ill-defined Fabs at the virion surface but in spite of the different resolution reported in Fig.2: 15.5Å for 5A and 21.8Å for 2C9 the latter seems much better defined with discrete density for the Fab.
- 6) Line 202: there is clear evidence of antigenic differences (especially for fusion loop antibodies) between the chimera particles (bYFV-17D – bYFVES504 – bYFVAsibi) but the same cannot be said concerning their structure. The bYFV17D preparation is more homogeneous and allowed for the structure to be solved but the fact that the authors were not able to resolve the structure of bYFES504 or bYFVAsibi at high resolution does not necessarily lead to the conclusion that the structure is therefore different (absence of evidence \neq evidence of absence). May be a softer formulation like “suggests” would be more appropriate.
- 7) Lines 209-211: the difference in morphology claimed in Fig.3 are not visible from the micrographs. Which other evidences prompted the authors to focus on the DIII mutants?
- 8) The effect of DIII swapping on the antigenicity of the chimera particle is more pronounced with the 17D backbone than with the Asibi or ES504 backbone. Can the authors comment on this result?
- 9) Lines 322-325 and Fig.5b: the claimed difference in morphology is not visible in the figure. Showing a single virion for the bYFV17D R380T mutant is not particular convincing, especially when this residue is so important for the conclusions of the paper. Can the authors please provide more images that clearly identify the “roughness” level of this important mutant?
- 10) Lines 377-379: again the “significant” effect of the mutation in position 380 is more evident in the context of 17D

backbone than in the context of ES504 backbone, suggesting a more complex correlation than the one described by the authors.

11) Did the authors try to solve the structure of bYFV17DR380T? Probably it might be impossible, and it might result in a structure similar to Figure 1 for bYFVES504, but that would be interesting by itself as circumstantial evidence indicating particle heterogeneity.

12) Lines 400-405: the first lines of the discussion seem a big overstatement. The structures of “virulent and vaccine strains” have not been solved. The surface lattice has been resolved on a chimera, which is significant, but these are not “authentic” virions of YFV.

13) In general, data on bYFVAsibi and its mutants chimera are mentioned but results are not shown.

Minor points:

- Please describe the rationale to construct the 17D/prM+TM mutant.

- Line 430: clarify here where 2C9 binds on the E protein.

- Line 433: it should be Supplementary Fig.10.

- Line 448: according to Fig. 6d it is E87 and not E67.

- Line 547: reference 34 does not describe the CPER method.

- Line 699: The test described is a FRNT and not a PRNT since an antibody is used to stain foci and no plaques are detected (see also line 713).

- Line 710: Which antibody is 4G4? Where does it bind?

Experimental notes:

1) Methods line 633: there is a mention of a “mature flavivirus” that is used as an in-house made template. This model seems to be used through all the 3D reconstructions to solve the structures. What is this template? How was it made? When was it used? Why were ab initio structures not used at all?

2) For the reconstructions both C1 or I3 symmetry are used but the rationale to use one or the other is not explained. Also why is I3 symmetry applied for icosahedral symmetry and not I1 for example?

3) Lines 170-183, Methods lines 641-650, Suppl. Fig.8, Suppl. Fig. 13: Please explain in significantly more detail the processing for the symmetry expansion pipeline. The methods are not detailed enough to understand the exact processing pipeline, and the reliance on a previous source does not give enough information to know exactly what is done in this case. For example, it is unclear if ab initio processes were performed on whole virions, or the symmetry expanded particles.

4) How the antigenic differences detected by the antibodies on the chimera correlate with the authentic viruses? The chimera may recapitulate the basic structure of the lattice, but they may not reflect the actual level of maturation of the authentic virions that is notably impacted by the action of nonstructural proteins.

5) A heavy emphasis of the paper is that the T380R mutation may drive a totally different structural phenotype which can be used as a platform for future flavivirus vaccine design. This is exciting, but the paper has shown only a single use-case (bYFVES504). A second confirmation, performing a T380R mutation and then structural characterization in the Asibi chimera would really drive home the message that this is a targetable pan-flavivirus (or at least pan-YFV) motif.

Figures and Tables:

1) Fig. 1 legend, line 146: What is meant by “at 4 degrees”? Is that the temperature to which the preparation was incubated before freezing? What is the rationale? Please clarify.

2) Fig. 1b: 17D class 3 and ES504 class 1 – it is not clear if these classes represent odd but consistent ‘squished’ particles’ or if they are averages of particles that are squished because they are near carbon or pressed against other virions. Is it possible to show some of the particles that make up this class? It is of interest whether this is a true subset of the virions or simply a sample preparation issue.

3) Suppl. Fig.2b: The particle does not look like it is 12.6 Å (cf with the 11.5Å structure from Suppl. Fig 5a) Indeed, the particle looks to be the same resolution as the ES504 particle from Fig 1. Particles in the 3D classification look to be aligned according to the imposed icosahedral symmetry much more than the particles themselves.

4) Suppl. Fig.3b: The only FSC curve provided is unmasked? Why? Please provide more detailed FSC measurements (masks included) to see whether this particle is really at that resolution.

5) Suppl. Fig. 8d: The residue labelled K96 is clearly a tyrosine. Also, the color of the numbering should be changed to black because it is barely visible.

6) Suppl. Fig. 9b: it would be beneficial to show the fitting of the epitope in the cryoEM density since only the free 2C9 structure was determined by X-ray crystallography.

7) Suppl. Fig.10: Which sequence is shown for YFV17D? It is not KF769015.1 used for the chimera since the latter has in pr a K6 and not, as shown in the figure, a E6. What’s the rationale of the color code for the highlighted AA? If it is yellow for same AA in 17D/Asibi and different in ES504 and orange for same AA in Asibi/ES504 and different in 17D please correct the color of AA S54/A54.

8) Suppl. Fig. 12: The second last processing stage depicted is basically a 20-angstrom ball, and then several steps are abbreviated as ‘3D refinement, post-processing and Bayesian polishing’ and then a 6.5 Å particle is presented. This seems an amazing increase in resolution given the inputs. Can you please present a step-by-step diagram of all steps of this pathway, including FCS curves for all the 3D classified particles, and any intermediaries presented along the way. Again, a clear tyrosine is labelled as K96 in Fig. 12e.

9) Suppl. Fig. 21: Could you highlight the differences between the two conformations in Fig. Suppl. 21d? This is not clear from the figure or the text.

10) Suppl. Table 2: Have the information on “other flavivirus vaccines” and “Pre-exposure to other flaviviruses” been used for analysis in this paper? If yes, what are the results? If not, they can be removed from the table.

11) Suppl. Table 4 and 5: Could the authors specify the pixel size at data collection and what type of binning (I assume they bin the data) they applied? Was the binning applied on the images or on the particles?

Reviewer #2

(Remarks to the Author)

Reviewer #3

(Remarks to the Author)

In this manuscript, Bibby and colleagues explore structural and antigenic differences among related strains of yellow fever virus (YFV). Despite the availability of an effective vaccine, YFV remains a global pathogen of concern due to sustained human infections and the potential for urban transmission. Here, the authors solve the structure of the YFV vaccine strain 17D using an innovative chimeric virus approach and cryo-EM. The resulting structural models were unremarkable compared to the known herringbone array of antiparallel dimers of mature virions. In contrast, and of great interest, a more representative circulating strain (ES504) appeared to have a different structure, as did the Asibi strain from which 17D was derived. Structural studies of ES504 and Asibi revealed a more heterogeneous structure the authors referred to as “bumpy.” Neutralization sensitivity to E protein fusion loop antibodies was a functional correlate of this alternative structure. In a stepwise series of mutagenesis studies, the authors identified a single amino acid substitution in domain III of E as a key residue involved in the herringbone arrangement of E proteins characteristic of YF17D and several other mature flavivirus structures.

This is an important paper that is well-crafted. Prior studies with dengue virus (DENV), West Nile virus (WNV), and YFV demonstrate that amino acid variation in the E protein may impact the antigenic structure of virions, similar to the neutralization studies presented in this paper. This is the first study to correlate changes in antigenicity/function and structure. I think the work will stimulate new experiments in the field.

The experiments missing from the paper are structural studies with a fusion loop-reactive antibody. The paper presents data with two monoclonal antibodies that neutralize all three strains tested (albeit to varying degrees). The key questions raised by the authors (and interesting potential impact on the vaccine field) relate to access to the fusion loop. While the fusion loop antibodies fail to neutralize YFV17D, how they engage the “bumpy” forms is of considerable interest and relevance. I suspect fusion loop mAbs also bind YF-17D in solution, too. Does mAb binding alter the arrangement of 17D E proteins to look more like Asibi?

How do the authors define virion stability (as related to line 428)? An appealing aspect of the paper is the connection between structure data and measures of virion function. The heterogeneity underlying the bumpy reconstructions may not correspond to virions with reduced infectivity over time (stability).

Version 1:

Reviewer comments:

Reviewer #1

(Remarks to the Author)

Comments to detailed Reviewer Responses (only the points with re-review are presented):

Reviewer 1 (and 2):

1. The title seems too strong and shall include the mention of the chimera system.

Whilst the chimeric approach allowed us to overcome previous barriers to resolving the structure of YFV particle surface, it is not the main focus of this manuscript and including reference to “chimera system” within the title would obscure the primary findings of this work. We already highlight the chimeric system in the abstract and in response to Reviewer 1 have added additional background in the introduction to help clarify the system to readers not familiar (noting that the BinJV chimera system has now appeared in 15 publications). We would like to also highlight that we have already robustly validated the chimeric approach for high resolution structural studies of flaviviruses which is published in Nature Communications (<https://www.nature.com/articles/s41467-021-22773-1>), and that this work is appropriately referenced in the current manuscript.

Although we prefer the succinct title currently provided, an alternative “A single residue in the yellow fever virus envelope protein modulates virion architecture and antigenicity” is also acceptable should the editor agree with this suggestion.

We disagree with the authors' statement “including reference to “chimera system” within the title would obscure the primary findings of this work”. We believe that the actual title is misleading, suggesting that authentic virus has been used while the system is based on a chimera virus. The findings are important and will not be diminished by the mention of the chimera system. If the authors are reluctant in having the wording “yellow fever/Binjari chimera” in the title, we find the alternative title

(focusing on the envelope protein) more fitting to the actual data presented.

3. The structural differences claimed in Fig.1 are not clearly visible from the micrographs or the 2D classes and the cryoEM reconstructions are not comparable in terms of resolution to appreciate a difference.

While we appreciate that qualitative assessment of the micrographs in Fig. 1 is up to individual interpretation, we can assure the reviewer that the bumpy and uneven surface seen for the ES504 and Asibi particles as well as the smooth 17D derived particles are representative of the full dataset. We provide some more example images from these data to support the representative images in Fig. 1. The additional micrographs have been added to Supplementary Fig. 1.

Further, these differences are actually reflected in the 2D classes, the vast majority of b17D particles (62.9%) are represented in a class that has a complete lipid bilayer and single E ectodomain outer layer (Fig1b, b17D class 4). In contrast, the majority of ES particles are represented in a class that does not have a complete density for either the lipid bilayer or E protein (class 3 - 80.1%), which is consistent with averaging particles that have a heterogeneous surface. Also note that processing of the bES504 class 4 (17.2%) which had slightly more defined features (but still less defined than the major b17D class) did not resolve a classically mature flavivirus structure (Sup Fig. 2b)

While this statement is true, the presented data support the heterogeneity of the wild type viruses and not their “bumpy” structure. The processing did not resolve any structure, probably due to the lower number of particles compared to the ones used to resolve the YF17D structure. I understand that the antibody binding data would suggest a bumpy structure for the wild type (to expose the FL) and this is probably the case, but, in my opinion, while the cryoEM data show a difference in the heterogeneity of the two viruses (17D vs ES504), they do not directly show the exposure of the fusion loop.

Finally, the particle differences are also reflected in the 3D refinements, and although the resolution is not the same, the features of the map are what is key. To demonstrate this, we provide the analysis below that shows that filtering the b17D to equivalent resolution to the bES504 map retains distinct E protein herringbone features that define a classical mature flavivirus (whilst the ES504 does not). Further filtering of each map to the resolution of the first WNV structure solved (DOI: 10.1126/science.1089316) shows how the b17D structure has similar surface features to mature WNV whilst the bES504 particle does not.

I think this figure could be a nice addition to the suppl. figures because it shows that the wild type is more heterogenous than 17D. The comparison with mature WNV is interesting because this virus has been shown to expose fusion loop epitopes (doi.org/10.4049/jimmunol.0900093) but the structure looks like 17D (no FL exposure).

4. There is no mention of why the bYFV-Asibi structure was not pursued/solved.

Initially a structure of bYFV-Asibi was not pursued due to limited beam-time on the microscopes. However, during the revision process, bYFV-Asibi has been imaged and included in Supplementary Fig. 22.

Micrographs of Asibi still appear in Fig. 1 without further going into detail in the text why analysis of these micrographs was not proceeded. This is confusing since Asibi appears again later in the manuscript (Fig 4)

10. Lines 377-379: again the “significant” effect of the mutation in position 380 is more evident in the context of 17D backbone than in the context of ES504 backbone, suggesting a more complex correlation than the one described by the authors.

We agree this is an interesting observation, but these results are consistent with our proposed role for this site within the viral particle context (i.e. this residue sits at inter-raft interfaces). Therefore, the level of the 380 effect would be intrinsically linked to the complex context of the viral strain, which differs at many more sites beyond residue 380. This therefore underscores the significance of our finding of reciprocal effects.

This point is not presented in the discussion but it might be relevant because the fact that the two mirror chimera do not behave the same it may imply that the “complex context of the viral strain” has a more relevant weight than the R380T mutation alone.

12. Lines 400-405: the first lines of the discussion seem a big overstatement. The structures of “virulent and vaccine strains” have not been solved. The surface lattice has been resolved on a chimera, which is significant, but these are not “authentic” virions of YFV.

The wording of line 400-405 has been adjusted to make it clear to readers that a chimeric platform was utilised to resolve these YFV virion structures. Furthermore, the platform itself is widely used and has been robustly demonstrated for the application in high resolution cryoEM studies, see our paper previously published in Nature Communications (<https://www.nature.com/articles/s41467-021-22773-1>).

We would complement the new added sentence as follows: “Here, we utilised a chimeric virus platform to reveal high resolution structures of vaccine and virulent envelope proteins of YFV”.

Experimental notes:

1. Methods line 633: there is a mention of a “mature flavivirus” that is used as an in-house made template. This model seems

to be used through all the 3D reconstructions to solve the structures. What is this template? How was it made? When was it used? Why were ab initio structures not used at all?

This construct was a low pass filtered map of a JEV group virus, and when filtered to 50Å did not have any residual primary structural (sequence level) information. Further, for all of the ASU refined maps, we first performed a reference free ab initio in cisTEM, which resulted in very similar icosahedral maps from both RELION and cisTEM pipelines.

The nature of the in-house template is not revealed in the material and methods section. Which virus of the JEV group was used?

4. How the antigenic differences detected by the antibodies on the chimera correlate with the authentic viruses? The chimera may recapitulate the basic structure of the lattice, but they may not reflect the actual level of maturation of the authentic virions that is notably impacted by the action of nonstructural proteins.

This is an important point. Please refer to our publication (DOI: 10.1126/scitranslmed.aax7888, Reference 34) which compares a large panel of antibodies for chimeras of DENV, WNV and ZIKV using this platform showing strong correlation with wild-type viruses. We also agree that particle maturity is important to address, and we have performed neutralisation assays using purified mature particles. These assays yield similar antibody sensitivity profiles to those obtained from unpurified virus, which may contain both mature and immature particles.

Has the enrichment in mature particles by purification been verified? While the tartrate gradient can separate flavivirus subviral particles from virus, I am not sure it can discriminate between immature and mature particles.

Figures and Tables:

3. Suppl. Fig.2b: The particle does not look like it is 12.6Å (cf with the 11.5Å structure from Suppl. Fig 5a) Indeed, the particle looks to be the same resolution as the ES504 particle from Fig 1. Particles in the 3D classification look to be aligned according to the imposed icosahedral symmetry much more than the particles themselves.

We thank the reviewer for spotting this typo. The refined particle with imposed icosahedral symmetry presented in Supplementary Fig. 2b is indeed the same resolution to the ES504 particle in Fig. 1 as it is the same map. We have amended the main text to 12.6 Å.

The reviewer is correct that it appears that there are imposed icosahedral features, which is why a C1 reconstruction was also implemented. However this did not reveal any more detailed information about the surface organisation of the ES504 particles. These findings suggest that E proteins on the ES504 particles can adopt different relative positions and the particles are inherently heterogeneous.

Why the number of particles in 2D classes is not shown in Suppl. Fig.2a but it is shown in Suppl. Fig.2b?

7. Suppl. Fig.10: Which sequence is shown for YFV17D? It is not KF769015.1 used for the chimera since the latter has in pr a K6 and not, as shown in the figure, a E6. What's the rationale of the color code for the highlighted AA? If it is yellow for same AA in 17D/Asibi and different in ES504 and orange for same AA in Asibi/ES504 and different in 17D please correct the color of AA S54/A54.

The coloring of amino acid 54 has been amended to the correct colour.

No response to the other questions has been provided. According to GenBank the sequence KF769015.1 has K6 (not E6) in pr and A54 (not S54) in E. Position 54 is then not different between 17D and wt and should not be coloured while position 56 (V in 17D and A in wt) should be coloured.

12.To allow for better comparison of 17D:2C9 and ES504-DIII:2C9 (Fig2b and Fig 4a), the same colouring should be used. In Fig2b: coloured by radial distance, Fig4a: coloured by local resolution. Fig 4a: same scale needed for local resolution to compare the particles better.

13.The ELISA binding data with selected antibodies (Suppl. Fig.4a) show less difference between 17D and wt viruses than the neutralization data presented in Fig.1e. It might be helpful to show a table with the EC50 of the ELISA data. Have the authors considered the effect of temperature on the different viruses tested? The neutralization assays were done at 28C, but the ELISA binding was performed at 37C. Would this affect the "bumpy" structure and the exposure of the FL? Since the reduction of "bumpiness" is proposed as a vaccine approach and vaccines will be acting at 37C, the relation temperature-structure is relevant.

Reviewer #2

(Remarks to the Author)

Reviewer #3

(Remarks to the Author)

This manuscript provides important new information about the structural ensembles that define the biology and function of

flaviviruses. My initial review of the paper was favorable because it adds new depth to our understanding of how changes in the sequence of viruses impact the shapes they assume, and the biological consequences of these differences. The authors addressed both points I raised during my review, and I appreciate the investment in exploring the mechanisms of access to the fusion loop. I note that the manuscript received numerous comments related to structural details. While the authors appear to have responded reasonably, I defer to those SMEs on the appropriateness of the revision in this regard. An appropriately strengthened manuscript will significantly advance the field.

Well done.

Version 2:

Reviewer comments:

Reviewer #1

(Remarks to the Author)

The authors have addressed the majority of our concerns. Few points remain not clearly explained in the "Result" section and only briefly mentioned in the "Discussion".

For example the rationale of freezing the viruses at 4C is not explained. A comparison between 4C and 37C for the binding of antibodies before freezing is presented and it seems that, differently from other flaviviruses, there is no correlation between binding and virus breathing. This is barely mentioned in the discussion but not clearly analyzed.

The discrepancy between ELISA and neutralization is not underscored in the "Results" section and justified in the "Discussion" section by the use of detergent in the assay which clearly affect the structure of the virion. Correlation between ELISA on whole virion and neutralization has been well documented for other flaviviruses such for example TBE. If the discrepancy shown in this paper is due to a technical limitation we wonder what is the relevance of the presented data.

Overall the data presented in this paper are new and we believe will have an impact on the field.

Reviewer #2

(Remarks to the Author)

Detailed Reviewer Responses:

Reviewer 1 (and 2):

Nevertheless, while the virus studied in this work is relevant, the authors put a too strong emphasis on the “whole virion structure of yellow fever virus” (lines 90-91) which obscures the fact that in reality the reported structures are chimeras that recapitulate the surface lattice of yellow fever virus. This overstatement is pursued through the entire paper.

We have revised the manuscript to further highlight that the advances made in this work have been made possible through the use of a chimeric virus platform, but, importantly, do match what is observed in wildtype virus where experimental validation is possible.

In particular:

1. The title seems too strong and shall include the mention of the chimera system.

Whilst the chimeric approach allowed us to overcome previous barriers to resolving the structure of YFV particle surface, it is not the main focus of this manuscript and including reference to “chimera system” within the title would obscure the primary findings of this work. We already highlight the chimeric system in the abstract and in response to Reviewer 1 have added additional background in the introduction to help clarify the system to readers not familiar (noting that the BinJV chimera system has now appeared in 15 publications). We would like to also highlight that we have already robustly validated the chimeric approach for high resolution structural studies of flaviviruses which is published in Nature Communications (<https://www.nature.com/articles/s41467-021-22773-1>), and that this work is appropriately referenced in the current manuscript.

Although we prefer the succinct title currently provided, an alternative “A single residue in the yellow fever virus envelope protein modulates virion architecture and antigenicity” is also acceptable should the editor agree with this suggestion.

2. The introduction would benefit from the presentation of the different yellow fever strains studied (17D, Asibi and ES504) and a workflow on how the Binjari system works.

A brief description of each YFV strain and the Binjari virus platform has been added to the introduction. A schematic depicting the genome design of the chimeric bYFV viruses has been added to Supplementary Fig. 1.

3. The structural differences claimed in Fig.1 are not clearly visible from the micrographs or the 2D classes and the cryoEM reconstructions are not comparable in terms of resolution to appreciate a difference.

While we appreciate that qualitative assessment of the micrographs in Fig. 1 is up to individual interpretation, we can assure the reviewer that the bumpy and uneven surface seen for the ES504 and Asibi particles as well as the smooth 17D derived particles are representative of the full dataset. We provide some more example images from these data to support the representative images in Fig. 1. The additional micrographs have been added to Supplementary Fig. 1.

Further, these differences are actually reflected in the 2D classes, the vast majority of b17D particles (62.9%) are represented in a class that has a complete lipid bilayer and single E ectodomain outer layer (Fig1b, b17D class 4). In contrast, the majority of ES particles are represented in a class that does not have a complete density for either the lipid bilayer or E protein (class 3 - 80.1%), which is consistent with averaging particles that have a heterogeneous surface. Also note that processing of the bES504 class 4 (17.2%) which had slightly more defined features (but still less

defined than the major b17D class) did not resolve a classically mature flavivirus structure (Sup Fig. 2b)

Finally, the particle differences are also reflected in the 3D refinements, and although the resolution is not the same, the features of the map are what is key. To demonstrate this, we provide the analysis below that shows that filtering the b17D to equivalent resolution to the bES504 map retains distinct E protein herringbone features that define a classical mature flavivirus (whilst the ES504 does not). Further filtering of each map to the resolution of the first WNV structure solved ([DOI: 10.1126/science.1089316](https://doi.org/10.1126/science.1089316)) shows how the b17D structure has similar surface features to mature WNV whilst the bES504 particle does not.

4. There is no mention of why the bYFV-Asibi structure was not pursued/solved. Initially a structure of bYFV-Asibi was not pursued due to limited beam-time on the microscopes. However, during the revision process, bYFV-Asibi has been imaged and included in Supplementary Fig. 22.
5. Lines 166-167: the density maps of bYFV-ES504 in complex with 5A or 2C9 antibody are described as incomplete and with ill-defined Fabs at the virion surface but in spite of the different resolution reported in Fig.2: 15.5Å for 5A and 21.8Å for 2C9 the latter seems much better defined with discrete density for the Fab.
We thank the authors for this comment. There was a typo in Line 163. bYFV-ES504:5A was resolved to a resolution of 21.8 Å. and bYFV-ES504:2C9 was resolved to 15.5 Å. This typo has been amended.
6. Line 202: there is clear evidence of antigenic differences (especially for fusion loop antibodies) between the chimera particles (bYFV-17D – bYFVES504 – bYFVAsibi) but the same cannot be said concerning their structure. The bYFV17D preparation is more homogeneous and allowed for the structure to be solved but the fact that the authors were not able to resolve the structure of bYFES504 or bYFVAsibi at high resolution does not necessarily lead to the conclusion that the structure is therefore different (absence of evidence \neq evidence of absence). Maybe a softer formulation like “suggests” would be more appropriate.
The wording of line 202 has been adjusted.

7. Lines 209-211: the difference in morphology claimed in Fig.3 are not visible from the micrographs. Which other evidences prompted the authors to focus on the DIII mutants?

We now present contrast enhanced images to enhance the definition of the viral particles presented. As we observed reciprocal effects for the DIII swaps (YFV 17D DIII in ES504 background the particles are visually smoother, and ES504 DIII in 17D background the particles are visually bumpier) this was sufficient information for us to proceed to examine individual mutants for DIII.

8. The effect of DIII swapping on the antigenicity of the chimera particle is more pronounced with the 17D backbone than with the Asibi or ES504 backbone. Can the authors comment on this result?

Please see the response to comment 10 below.

9. Lines 322-325 and Fig.5b: the claimed difference in morphology is not visible in the figure. Showing a single virion for the bYFV17D R380T mutant is not particularly convincing, especially when this residue is so important for the conclusions of the paper. Can the authors please provide more images that clearly identify the “roughness” level of this important mutant?

To confirm reproducibility, a new preparation of the bYFV_{17D} R380T mutant virus has been purified and imaged via cryo-EM. Fig. 5 has been updated with a new representative micrograph.

10. Lines 377-379: again the “significant” effect of the mutation in position 380 is more evident in the context of 17D backbone than in the context of ES504 backbone, suggesting a more complex correlation than the one described by the authors.

We agree this is an interesting observation, but these results are consistent with our proposed role for this site within the viral particle context (i.e. this residue sits at inter-raft interfaces). Therefore, the level of the 380 effect would be intrinsically linked to the complex context of the viral strain, which differs at many more sites beyond residue 380. This therefore underscores the significance of our finding of reciprocal effects.

11. Did the authors try to solve the structure of bYFV-17D R380T? Probably it might be impossible, and it might result in a structure similar to Fig. 1 for bYFV-ES504, but that would be interesting by itself as circumstantial evidence indicating particle heterogeneity.

Due to limited beam time on the microscope, we prioritised imaging bAsibi and bAsibi T380R as recommended by the editor.

12. Lines 400-405: the first lines of the discussion seem a big overstatement. The structures of “virulent and vaccine strains” have not been solved. The surface lattice has been resolved on a chimera, which is significant, but these are not “authentic” virions of YFV.

The wording of line 400-405 has been adjusted to make it clear to readers that a chimeric platform was utilised to resolve these YFV virion structures. Furthermore, the platform itself is widely used and has been robustly demonstrated for the application in high resolution cryoEM studies, see our paper previously published in Nature Communications (<https://www.nature.com/articles/s41467-021-22773-1>).

13. In general, data on bYFVAsibi and its mutants chimera are mentioned but results are not shown.

For clarity we have added representative micrographs of the bAsibi/DIII mutant chimeras into Supplementary Fig. 14. All other results in relation to the bYFV-Asibi chimeras are shown in Fig. 1e (neutralisation profile of bYFV-Asibi), Fig. 4e (structure of bYFV-Asibi/DIII-17D), Fig. 4g (neutralisation profile of the bYFV-Asibi/DIII chimeras).

Minor points:

- Please describe the rationale to construct the 17D/prM+TM mutant.

This construct was designed in order to assess if there was a synergistic phenotype that required the transmembrane regions of both E and M (which are closely associated in mature flavivirus virion structures).

- Line 430: clarify here where 2C9 binds on the E protein.
This information has been added into line 430.

- Line 433: it should be Supplementary Fig.10.
This error has been fixed.

- Line 448: according to Fig. 6d it is E87 and not E67.
This typo in line 448 has been corrected.

- Line 547: reference 34 does not describe the CPER method.
The authors would like to thank the reviewers for bringing this to our attention. The correct reference, Piyasena et al. 2017, has now been added to the manuscript.

- Line 699: The test described is a FRNT and not a PRNT since an antibody is used to stain foci and no plaques are detected (see also line 713).
PRNT has been corrected to FRNT throughout the manuscript.

- Line 710: Which antibody is 4G4? Where does it bind?
4G4 is an anti-flavivirus NS1 antibody. This has been added into methods section.

Experimental notes:

1. Methods line 633: there is a mention of a “mature flavivirus” that is used as an in-house made template. This model seems to be used through all the 3D reconstructions to solve the structures. What is this template? How was it made? When was it used? Why were *ab initio* structures not used at all?

This construct was a low pass filtered map of a JEV group virus, and when filtered to 50Å did not have any residual primary structural (sequence level) information. Further, for all of the ASU refined maps, we first performed a reference free *ab initio* in cisTEM, which resulted in very similar icosahedral maps from both RELION and cisTEM pipelines.

2. For the reconstructions both C1 or I3 symmetry are used but the rationale to use one or the other is not explained. Also why is I3 symmetry applied for icosahedral symmetry and not I1 for example?

C1 symmetry can be considered to have no specific symmetry imposed and was used to assess if imposing icosahedral symmetry was playing a large role in the different resolutions and map features that we observed between the 17D and ES504

derived particles. I3 and I1 are both icosahedral symmetry but are oriented differently in coordinate space, resulting in different positioning of their symmetry axes during refinement. These should and do produce very similar maps, but we used I1 for cisTEM processing as it allowed straightforward symmetry expansion using a previously established script (Reference 41).

3. Lines 170-183, Methods lines 641-650, Suppl. Fig.8, Suppl. Fig. 13: Please explain in significantly more detail the processing for the symmetry expansion pipeline. The methods are not detailed enough to understand the exact processing pipeline, and the reliance on a previous source does not give enough information to know exactly what is done in this case. For example, it is unclear if ab initio processes were performed on whole virions, or the symmetry expanded particles.

More details regarding the symmetry expansion pipeline have been added to the Image Processing and Single-Particle Analysis methods section. For the datasets processed in cisTEM2, the ab initio was performed on the whole virions. This has been clarified in text.

4. How the antigenic differences detected by the antibodies on the chimera correlate with the authentic viruses? The chimera may recapitulate the basic structure of the lattice, but they may not reflect the actual level of maturation of the authentic virions that is notably impacted by the action of nonstructural proteins.

This is an important point. Please refer to our publication (DOI: [10.1126/scitranslmed.aax7888](https://doi.org/10.1126/scitranslmed.aax7888), Reference 34) which compares a large panel of antibodies for chimeras of DENV, WNV and ZIKV using this platform showing strong correlation with wild-type viruses. We also agree that particle maturity is important to address, and we have performed neutralisation assays using purified mature particles. These assays yield similar antibody sensitivity profiles to those obtained from unpurified virus, which may contain both mature and immature particles.

5. A heavy emphasis of the paper is that the T380R mutation may drive a totally different structural phenotype which can be used as a platform for future flavivirus vaccine design. This is exciting, but the paper has shown only a single use-case (bYFVES504). A second confirmation, performing a T380R mutation and then structural characterization in the Asibi chimera would really drive home the message that this is a targetable pan-flavivirus (or at least pan-YFV) motif.

We recognise this is an important point but also that this is not trivial from an experimental standpoint. However, given the significance of this for the field we generated a new chimera bYFV-Asibi T380R and now report the structure of this particle compared to bYFV-Asibi as well as neutralisation assays. Due to microscope downtime on our flagship 300 kV cryo-EM microscope, we collected both bYFV-Asibi and bYFV-Asibi T380R SPA datasets on a 200 kV instrument, leading to a slight reduction in map resolution. However, the results are clear and we show that bYFV-Asibi T380R adopts a classical mature flavivirus structure and that it is not as sensitive to fusion loop mAb neutralisation, whilst no clear surface E features are observed for bYFV-Asibi and it is sensitive to fusion loop mAb neutralisation. These findings are in agreement with our hypothesis that R380 offers a generalisable simple solution to stabilise the mature particle conformation of diverse YFV strains. This is summarised in a new figure (Supplementary Fig. 22).

Figures and Tables:

1. Fig. 1 legend, line 146: What is meant by “at 4 degrees”? Is that the temperature to which the preparation was incubated before freezing? What is the rationale? Please clarify.

The virus was incubated at 4°C prior to freezing and this has been clarified in the figure legend.

2. Fig. 1b: 17D class 3 and ES504 class 1 – it is not clear if these classes represent odd but consistent ‘squished’ particles’ or if they are averages of particles that are squished because they are near carbon or pressed against other virions. Is it possible to show some of the particles that make up this class? It is of interest whether this is a true subset of the virions or simply a sample preparation issue.

These “squished” particles are observed pressed up to other virions as well as in isolation away from other particles. We have attached some particle images of bYFV-17D which illustrates this.

3. Suppl. Fig.2b: The particle does not look like it is 12.6Å (cf with the 11.5Å structure from Suppl. Fig 5a) Indeed, the particle looks to be the same resolution as the ES504 particle from Fig 1. Particles in the 3D classification look to be aligned according to the imposed icosahedral symmetry much more than the particles themselves.

We thank the reviewer for spotting this typo. The refined particle with imposed icosahedral symmetry presented in Supplementary Fig. 2b is indeed the same resolution to the ES504 particle in Fig. 1 as it is the same map. We have amended the main text to 12.6 Å.

The reviewer is correct that it appears that there are imposed icosahedral features, which is why a C1 reconstruction was also implemented. However this did not reveal any more detailed information about the surface organisation of the ES504 particles. These findings suggest that E proteins on the ES504 particles can adopt different relative positions and the particles are inherently heterogeneous.

4. Suppl. Fig.3b: The only FSC curve provided is unmasked? Why? Please provide more detailed FSC measurements (masks included) to see whether this particle is really at that resolution.

We show unmasked resolutions as this provides an unbiased assessment of the global (or nominal) resolution of a given map. If an overly tight shaped mask is applied, the mask itself can contribute towards the resolution and can result in an inflated claimed resolution. We also include a local resolution map in Supplementary Fig. 3, where it can be seen that alignment of the lipid bilayer (which is roughly spherical at this resolution) is contributing to the unmasked FSC resolution. As there is no clear E density there is no rationale to masking and further FSC analysis for this map.

5. Suppl. Fig. 8d: The residue labeled K96 is a tyrosine. Also, the color of the numbering should be changed to black because it is barely visible.

The label for amino acid 96 has been amended.

6. Suppl. Fig. 9b: it would be beneficial to show the fitting of the epitope in the cryoEM density since only the free 2C9 structure was determined by X-ray crystallography. Supplementary Fig. 9 has been amended to include a new panel with the model of bYFV_{17D}:2C9 in the cryo-EM density map.
7. Suppl. Fig. 10: Which sequence is shown for YFV17D? It is not KF769015.1 used for the chimera since the latter has in pr a K6 and not, as shown in the figure, a E6. What's the rationale of the color code for the highlighted AA? If it is yellow for same AA in 17D/Asibi and different in ES504 and orange for same AA in Asibi/ES504 and different in 17D please correct the color of AA S54/A54. The coloring of amino acid 54 has been amended to the correct colour.
8. Suppl. Fig. 12: The second last processing stage depicted is basically a 20-angstrom ball, and then several steps are abbreviated as '3D refinement, post-processing and Bayesian polishing' and then a 6.5 Å particle is presented. This seems an amazing increase in resolution given the inputs. Can you please present a step-by-step diagram of all steps of this pathway, including FCS curves for all the 3D classified particles, and any intermediaries presented along the way. Again, a clear tyrosine is labelled as K96 in Fig. 12e. Supplementary Fig. 12 has been updated to include several additional processing steps to help clarify the SPA pathway. Regarding FSC requests for 3D classification steps, Relion 3D classification does not directly output Fourier Shell Correlation (FSC) curves as half-maps are not generated during this process. Additionally, the mislabelled tyrosine has also been amended, we thank the reviewer for picking up this mistake.
9. Suppl. Fig. 21: Could you highlight the differences between the two conformations in Fig. Suppl. 21d? This is not clear from the figure or the text. A new panel has been added to Supplementary Fig. 21 which highlights the differences between the two confirmations.
10. Suppl. Table 2: Have the information on "other flavivirus vaccines" and "Pre-exposure to other flaviviruses" been used for analysis in this paper? If yes, what are the results? If not, they can be removed from the table. The "Other flavivirus vaccines" and "Pre-exposure to other flaviviruses" columns have been removed from Supplementary Table 2.
11. Suppl. Table 4 and 5: Could the authors specify the pixel size at data collection and what type of binning (I assume they bin the data) they applied? Was the binning applied on the images or on the particles? The pixel size at data collection is already stated in Supplementary Tables 4 and 5 as the "Nominal Pixel Size". The images were binned during processing, this information has been added into the Image Processing and Single-Particle Analysis methods section.

Reviewer 3:

1. The experiments missing from the paper are structural studies with a fusion loop-reactive antibody. The paper presents data with two monoclonal antibodies that neutralize all three strains tested (albeit to varying degrees). The key questions raised by the authors (and interesting potential impact on the vaccine field) relate to access to the fusion loop. While the fusion loop antibodies fail to neutralize YFV17D, how they engage the “bumpy” forms is of considerable interest and relevance. I suspect fusion loop mAbs also bind YF-17D in solution, too. Does mAb binding alter the arrangement of 17D E proteins to look more like Asibi?

We agree with the reviewer that this is an important research question. In response, we have performed complexing and imaging of fusion loop specific Fab (2A10G6) together with 17D and ES504 derived particles (shown in Supplemental Fig. 4). To our knowledge this is the first such study for any mature flavivirus. Full SPA datasets were not collected due to particle aggregation. However, sufficient micrographs of non-aggregated particles were collected to allow qualitative assessment. Interestingly, we find that the 2A10G6 Fab appears to bind well to ES504 derived particles but only a few 2A10G6 Fab like densities can be seen for 17D preparations (at matched concentration). In contrast, the DIII reactive 17D-specific Fab 864 complexed well with 17D particles but not ES504. We did not proceed to SPA analysis with the fusion loop Fab as it is expected that the particle heterogeneity will prevent refinement to high resolution using icosahedral based approaches (noting that there have been no structures of mature particles solved with a fusion loop antibody bound for any flavivirus, likely due to this same limitation). Nevertheless, this visualisation is important to understand our observed neutralisation profiles in the context of cryo-EM and we thank the reviewer for this suggestion.

2. How do the authors define virion stability (as related to line 428)? An appealing aspect of the paper is the connection between structure data and measures of virion function. The heterogeneity underlying the bumpy reconstructions may not correspond to virions with reduced infectivity over time (stability).

We thank the reviewer for picking up on this point. We acknowledge that without definition use of stability may be confusing. We do not refer to shelf stability (e.g. half-life) of the particles but are referring to rigidity. Our findings are consistent with a model where the R380 mutation provides additional inter-raft contacts in the “classical” mature flavivirus herringbone arrangement, which lock (or stabilise) this conformation. We have altered some of the discussion to highlight this important point.

Comments to detailed Reviewer Responses:

Reviewer 1 (and 2):

Nevertheless, while the virus studied in this work is relevant, the authors put a too strong emphasis on the “whole virion structure of yellow fever virus” (lines 90-91) which obscures the fact that in reality the reported structures are chimeras that recapitulate the surface lattice of yellow fever virus. This overstatement is pursued through the entire paper.

We have revised the manuscript to further highlight that the advances made in this work have been made possible through the use of a chimeric virus platform, but, importantly, do match what is observed in wildtype virus where experimental validation is possible.

In particular:

1. The title seems too strong and shall include the mention of the chimera system.

Whilst the chimeric approach allowed us to overcome previous barriers to resolving the structure of YFV particle surface, it is not the main focus of this manuscript and including reference to “chimera system” within the title would obscure the primary findings of this work. We already highlight the chimeric system in the abstract and in response to Reviewer 1 have added additional background in the introduction to help clarify the system to readers not familiar (noting that the BinJV chimera system has now appeared in 15 publications). We would like to also highlight that we have already robustly validated the chimeric approach for high resolution structural studies of flaviviruses which is published in Nature Communications (<https://www.nature.com/articles/s41467-021-22773-1>), and that this work is appropriately referenced in the current manuscript.

Although we prefer the succinct title currently provided, an alternative “A single residue in the yellow fever virus envelope protein modulates virion architecture and antigenicity” is also acceptable should the editor agree with this suggestion.

We disagree with the authors’ statement “including reference to “chimera system” within the title would obscure the primary findings of this work”. We believe that the actual title is misleading, suggesting that authentic virus has been used while the system is based on a chimera virus. The findings are important and will not be diminished by the mention of the chimera system. If the authors are reluctant in having the wording “yellow fever/Binjari chimera” in the title, we find the alternative title (focusing on the envelope protein) more fitting to the actual data presented.

We have opted to use the alternative title “A single residue in the yellow fever virus envelope protein modulates virion architecture and antigenicity”.

2. The introduction would benefit from the presentation of the different yellow fever strains studied (17D, Asibi and ES504) and a workflow on how the Binjari system works.

A brief description of each YFV strain and the Binjari virus platform has been added to the introduction. A schematic depicting the genome design of the chimeric bYFV viruses has been added to Supplementary Fig. 1.

3. The structural differences claimed in Fig.1 are not clearly visible from the micrographs or the 2D classes and the cryo-EM reconstructions are not comparable in terms of resolution to appreciate a difference.

While we appreciate that qualitative assessment of the micrographs in Fig. 1 is up to individual interpretation, we can assure the reviewer that the bumpy and uneven surface seen for the ES504 and Asibi particles as well as the smooth 17D derived particles are representative of the full dataset. We provide some more example images from these data to support the representative images in Fig. 1. The additional micrographs have been added to Supplementary Fig. 1.

Further, these differences are actually reflected in the 2D classes, the vast majority of b17D particles (62.9%) are represented in a class that has a complete lipid bilayer and single E ectodomain outer layer (Fig1b, b17D class 4). In contrast, the majority of ES particles are represented in a class that does not have a complete density for either the lipid bilayer or E protein (class 3 - 80.1%), which is consistent with averaging particles that have a heterogeneous surface. Also note that processing of the bES504 class 4 (17.2%) which had slightly more defined features (but still less defined than the major b17D class) did not resolve a classically mature flavivirus structure (Sup Fig. 2b)

While this statement is true, the presented data support the heterogeneity of the wild type viruses and not their “bumpy” structure. The processing did not resolve any structure, probably due to the lower number of particles compared to the ones used to resolve the YFV-17D structure.

There were ~18,000 particles used to resolve the bYFV_{ES504} map and only ~3,300 particles used to resolve the bYFV_{17D} map. This information is available in Supplemental Fig.2.

I understand that the antibody binding data would suggest a bumpy structure for the wild type (to expose the FL) and this is probably the case, but, in my opinion, while the cryo-EM data shows a difference in the heterogeneity of the two viruses (17D vs ES504), they do not directly show the exposure of the fusion loop.

We appreciate this point and would like to highlight that our heterogenous structures indicate that the virulent sequence derived virions are not in the classical icosahedral conformation and together with the visualisation of FLE Fab bound bYFV_{ES504} virions by cryo-EM, as well as our neutralisation results, the most parsimonious explanation is that the FLE is exposed on the bYFV_{ES504} virions.

Finally, the particle differences are also reflected in the 3D refinements, and although the resolution is not the same, the features of the map are what is key. To demonstrate this, we provide the analysis below that shows that filtering the b17D to equivalent resolution to the bES504 map retains distinct E protein herringbone features that define a classical mature flavivirus (whilst the ES504 does not). Further filtering of each map to the resolution of the first WNV structure solved ([DOI: 10.1126/science.1089316](https://doi.org/10.1126/science.1089316)) shows how the b17D structure has similar surface features to mature WNV whilst the bES504 particle does not.

I think this figure could be a nice addition to the suppl. figures because it shows that the wild type is more heterogenous than 17D. The comparison with mature WNV is interesting because this virus has been shown to expose fusion loop epitopes (doi.org/10.4049/jimmunol.0900093) but the structure looks like 17D (no FL exposure).

This figure has been added to Supplementary Figure 3. Although there is some reports of modest neutralisation of FLE antibodies targeting WNV, the FLE is thought to be poorly exposed in the mature virion conformation (see <https://doi.org/10.1128/jvi.05859-11>).

- There is no mention of why the bYFV-Asibi structure was not pursued/solved. Initially a structure of bYFV-Asibi was not pursued due to limited beam-time on the microscopes. However, during the revision process, bYFV-Asibi has been imaged and included in Supplementary Fig. 22.

Micrographs of Asibi still appear in Fig. 1 without further going into detail in the text why analysis of these micrographs was not proceeded. This is confusing since Asibi appears again later in the manuscript (Fig 4).

The wording of the first results paragraph has been reworded to reduce confusion regarding the chronology of results.

- Lines 166-167: the density maps of bYFV-ES504 in complex with 5A or 2C9 antibody are described as incomplete and with ill-defined Fabs at the virion surface but in spite of the different resolution reported in Fig.2: 15.5Å for 5A and 21.8Å for 2C9 the latter seems much better defined with discrete density for the Fab. We thank the authors for this comment. There was a typo in Line 163. bYFVES504:5A was resolved to a resolution of 21.8 Å. and bYFV-ES504:2C9 was resolved to 15.5 Å. This typo has been amended.

6. Line 202: there is clear evidence of antigenic differences (especially for fusion loop antibodies) between the chimera particles (bYFV-17D – bYFVES504 – bYFVAsibi) but the same cannot be said concerning their structure. The bYFV17D preparation is more homogeneous and allowed for the structure to be solved but the fact that the authors were not able to resolve the structure of bYFES504 or bYFVAsibi at high resolution does not necessarily lead to the conclusion that the structure is therefore different (absence of evidence \neq evidence of absence). Maybe a softer formulation like “suggests” would be more appropriate.

The wording of line 202 has been adjusted.

7. Lines 209-211: the difference in morphology claimed in Fig.3 are not visible from the micrographs. Which other evidences prompted the authors to focus on the DIII mutants?

We now present contrast enhanced images to enhance the definition of the viral particles presented. As we observed reciprocal effects for the DIII swaps (YFV 17D DIII in ES504 background the particles are visually smoother, and ES504 DIII in 17D background the particles are visually bumpier) this was sufficient information for us to proceed to examine individual mutants for DIII.

8. The effect of DIII swapping on the antigenicity of the chimera particle is more pronounced with the 17D backbone than with the Asibi or ES504 backbone. Can the authors comment on this result?

Please see the response to comment 10 below.

9. Lines 322-325 and Fig.5b: the claimed difference in morphology is not visible in the figure. Showing a single virion for the bYFV17D R380T mutant is not particularly convincing, especially when this residue is so important for the conclusions of the paper. Can the authors please provide more images that clearly identify the “roughness” level of this important mutant?

To confirm reproducibility, a new preparation of the bYFV_{17D} R380T mutant virus has been purified and imaged via cryo-EM. Fig. 5 has been updated with a new representative micrograph.

10. Lines 377-379: again the “significant” effect of the mutation in position 380 is more evident in the context of 17D backbone than in the context of ES504 backbone, suggesting a more complex correlation than the one described by the authors.

We agree this is an interesting observation, but these results are consistent with our proposed role for this site within the viral particle context (i.e. this residue sits at inter-raft interfaces). Therefore, the level of the 380 effect would be intrinsically linked to the complex context of the viral strain, which differs at many more sites beyond residue 380. This therefore underscores the significance of our finding of reciprocal effects.

This point is not presented in the discussion however, it might be relevant because the fact that the mirror chimera does not behave the same, may imply that the “complex context of the viral strain” has a more relevant weight than the R380T mutation alone.

We have added the following sentence to highlight this point in the discussion. “Interestingly, although we observed mirrored reciprocity of the neutralisation sensitivity between vaccine and virulent strains when swapping residue 380, the magnitude of the effect was different, suggesting that other residues or combinations also play a role in FLE exposure.”

11. Did the authors try to solve the structure of bYFV-17D R380T? Probably it might be impossible, and it might result in a structure similar to Fig. 1 for bYFV-ES504, but that would be interesting by itself as circumstantial evidence indicating particle heterogeneity.

Due to limited beam time on the microscope, we prioritised imaging bAsibi and bAsibi T380R as recommended by the editor.

12. Lines 400-405: the first lines of the discussion seem a big overstatement. The structures of “virulent and vaccine strains” have not been solved. The surface lattice has been resolved on a chimera, which is significant, but these are not “authentic” virions of YFV.

The wording of line 400-405 has been adjusted to make it clear to readers that a chimeric platform was utilised to resolve these YFV virion structures. Furthermore, the platform itself is widely used and has been robustly demonstrated for the application in high resolution cryoEM studies, see our paper previously published in Nature Communications (<https://www.nature.com/articles/s41467-021-22773-1>).

We would complement the new added sentence as follows: “Here, we utilised a chimeric virus platform to reveal high resolution structures of vaccine and virulent envelope proteins of YFV”.

The wording of line 400-405 has been adjusted to include the reviewer’s suggestion.

13. In general, data on bYFVAsibi and its mutants chimera are mentioned but results are not shown.

For clarity we have added representative micrographs of the bAsibi/DIII mutant chimeras into Supplementary Fig. 14. All other results in relation to the bYFV-Asibi chimeras are shown in Fig. 1e (neutralisation profile of bYFV-Asibi), Fig. 4e (structure of bYFV-Asibi/DIII-17D), Fig. 4g (neutralisation profile of the bYFV-Asibi/DIII chimeras).

Minor points:

- Please describe the rationale to construct the 17D/prM+TM mutant.

This construct was designed in order to assess if there was a synergistic phenotype that required the transmembrane regions of both E and M (which are closely associated in mature flavivirus virion structures).

- Line 430: clarify here where 2C9 binds on the E protein.

This information has been added into line 430.

- Line 433: it should be Supplementary Fig.10.
This error has been fixed.
- Line 448: according to Fig. 6d it is E87 and not E67.
This typo in line 448 has been corrected.
- Line 547: reference 34 does not describe the CPER method.
The authors would like to thank the reviewers for bringing this to our attention. The correct reference, Piyasena et al. 2017, has now been added to the manuscript.
- Line 699: The test described is a FRNT and not a PRNT since an antibody is used to stain foci and no plaques are detected (see also line 713).
PRNT has been corrected to FRNT throughout the manuscript.
- Line 710: Which antibody is 4G4? Where does it bind?
4G4 is an anti-flavivirus NS1 antibody. This has been added into methods section.

Experimental notes:

1. Methods line 633: there is a mention of a “mature flavivirus” that is used as an inhouse made template. This model seems to be used through all the 3D reconstructions to solve the structures. What is this template? How was it made? When was it used? Why were *ab initio* structures not used at all?

This construct was a low pass filtered map of a JEV group virus, and when filtered to 50A did not have any residual primary structural (sequence level) information. Further, for all of the ASU refined maps, we first performed a reference free *ab initio* in cisTEM, which resulted in very similar icosahedral maps from both RELION and cisTEM pipelines.

The nature of the in-house template is not revealed in the material and methods section. Which virus of the JEV group was used?

The in-house generated reconstruction used throughout the manuscript is an unpublished structure of a Rocio virus chimera (bROCV). This detail has been added to the material and methods section.

2. For the reconstructions both C1 or I3 symmetry are used but the rationale to use one or the other is not explained. Also why is I3 symmetry applied for icosahedral symmetry and not I1 for example?

C1 symmetry can be considered to have no specific symmetry imposed and was used to assess if imposing icosahedral symmetry was playing a large role in the different resolutions and map features that we observed between the 17D and ES504 derived particles. I3 and I1 are both icosahedral symmetry but are oriented differently in coordinate space, resulting in different positioning of their symmetry axes during refinement. These should and do produce very similar maps, but we used I1 for cisTEM processing as it allowed straightforward symmetry expansion using a previously established script (Reference 41).

3. Lines 170-183, Methods lines 641-650, Suppl. Fig.8, Suppl. Fig. 13: Please explain in significantly more detail the processing for the symmetry expansion pipeline. The methods are not detailed enough to understand the exact processing pipeline, and the reliance on a previous source does not give enough information to know exactly what is done in this case. For example, it is unclear if ab initio processes were performed on whole virions, or the symmetry expanded particles.

More details regarding the symmetry expansion pipeline have been added to the Image Processing and Single-Particle Analysis methods section. For the datasets processed in cisTEM2, the ab initio was performed on the whole virions. This has been clarified in text.

4. How the antigenic differences detected by the antibodies on the chimera correlate with the authentic viruses? The chimera may recapitulate the basic structure of the lattice, but they may not reflect the actual level of maturation of the authentic virions that is notably impacted by the action of non-structural proteins.

This is an important point. Please refer to our publication (DOI: [10.1126/scitranslmed.aax7888](https://doi.org/10.1126/scitranslmed.aax7888), Reference 34) which compares a large panel of antibodies for chimeras of DENV, WNV and ZIKV using this platform showing strong correlation with wild-type viruses. We also agree that particle maturity is important to address, and we have performed neutralisation assays using purified mature particles. These assays yield similar antibody sensitivity profiles to those obtained from unpurified virus, which may contain both mature and immature particles.

Has the enrichment in mature particles by purification been verified? While the tartrate gradient can separate flavivirus subviral particles from virus, I am not sure it can discriminate between immature and mature particles.

The authors appreciate the reviewers' concerns, however, tartrate gradients are able to separate fully mature particles from partially mature particles, as shown in the example below and in the following publication: <https://doi.org/10.1038/s41541-024-00903-2>. Additionally, we would like to ensure the reviewers that the maturity of the purified virus preparation used for the mature virus neutralisation assay was verified via SDS-PAGE prior to use.

5. A heavy emphasis of the paper is that the T380R mutation may drive a totally different structural phenotype which can be used as a platform for future flavivirus vaccine design. This is exciting, but the paper has shown only a single use-case (bYFVES504). A second confirmation, performing a T380R mutation and then structural characterization in the Asibi chimera would really drive home the message that this is a targetable pan-flavivirus (or at least pan-YFV) motif.

We recognise this is an important point but also that this is not trivial from an experimental standpoint. However, given the significance of this for the field we generated a new chimera bYFV-Asibi T380R and now report the structure of this particle compared to bYFV-Asibi as well as neutralisation assays. Due to microscope downtime on our flagship 300 kV cryo-EM microscope, we collected both bYFV-Asibi and bYFV-Asibi T380R SPA datasets on a 200 kV instrument, leading to a slight reduction in map resolution. However, the results are clear and we show that bYFVAsibi T380R adopts a classical mature flavivirus structure and that it is not as sensitive to fusion loop mAb neutralisation, whilst no clear surface E features are observed for bYFV-Asibi and it is sensitive to fusion loop mAb neutralisation. These findings are in agreement with our hypothesis that R380 offers a generalisable simple solution to stabilise the mature particle conformation of diverse YFV strains. This is summarised in a new figure (Supplementary Fig. 22).

Figures and Tables:

1. Fig. 1 legend, line 146: What is meant by “at 4 degrees”? Is that the temperature to which the preparation was incubated before freezing? What is the rationale? Please clarify.

The virus was incubated at 4°C prior to freezing and this has been clarified in the figure legend.

2. Fig. 1b: 17D class 3 and ES504 class 1 – it is not clear if these classes represent odd but consistent ‘squished’ particles’ or if they are averages of particles that are squished because they are near carbon or pressed against other virions. Is it possible to show some of the particles that make up this class? It is of interest whether this is a true subset of the virions or simply a sample preparation issue.

These “squished” particles are observed pressed up to other virions as well as in isolation away from other particles. We have attached some particle images of bYFV17D which illustrates this.

3. Suppl. Fig.2b: The particle does not look like it is 12.6Å (cf with the 11.5Å structure from Suppl. Fig 5a) Indeed, the particle looks to be the same resolution as the ES504 particle from Fig 1. Particles in the 3D classification look to be aligned according to the imposed icosahedral symmetry much more than the particles themselves.

We thank the reviewer for spotting this typo. The refined particle with imposed icosahedral symmetry presented in Supplementary Fig. 2b is indeed the same resolution to the ES504 particle in Fig. 1 as it is the same map. We have amended the main text to 12.6 Å.

The reviewer is correct that it appears that there are imposed icosahedral features, which is why a C1 reconstruction was also implemented. However, this did not reveal any more detailed information about the surface organisation of the ES504 particles. These findings suggest that E proteins on the ES504 particles can adopt different relative positions and the particles are inherently heterogeneous.

Why the number of particles in 2D classes is not shown in Suppl. Fig.2a but it is shown in Suppl. Fig.2b?

This detail was initially overlooked and has now been added to Supplementary Figure 2a.

4. Suppl. Fig.3b: The only FSC curve provided is unmasked? Why? Please provide more detailed FSC measurements (masks included) to see whether this particle is really at that resolution.

We show unmasked resolutions as this provides an unbiased assessment of the global (or nominal) resolution of a given map. If an overly tight shaped mask is applied, the mask itself can contribute towards the resolution and can result in an inflated claimed resolution. We also include a local resolution map in Supplementary Fig. 3, where it can be seen that alignment of the lipid bilayer (which is roughly spherical at this resolution) is contributing to the unmasked FSC resolution. As there is no clear E density there is no rationale to masking and further FSC analysis for this map.

5. Suppl. Fig. 8d: The residue labeled K96 is a tyrosine. Also, the color of the numbering should be changed to black because it is barely visible. The label for amino acid 96 has been amended.

6. Suppl. Fig. 9b: it would be beneficial to show the fitting of the epitope in the cryoEM density since only the free 2C9 structure was determined by X-ray crystallography. Supplementary Fig. 9 has been amended to include a new panel with the model of bYFV_{17D}:2C9 in the cryo-EM density map.

7. Suppl. Fig.10: Which sequence is shown for YFV17D? It is not KF769015.1 used for the chimera since the latter has in pr a K6 and not, as shown in the figure, a E6. What's the rationale of the colour code for the highlighted AA? If it is yellow for same AA in 17D/Asibi and different in ES504 and orange for same AA in Asibi/ES504 and different in 17D please correct the colour of AA S54/A54.

The colouring of amino acid 54 has been amended to the correct colour.

No response to the other questions has been given. According to GenBank the sequence KF769015.1 has K6 (not E6) in pr and A54 (not S54) in E. Position 54 is

then not different between 17D and wt and should not be coloured while position 56 (V in 17D and A in wt) should be coloured.

The errors in the YFV-17D amino acid sequence and the colouring of the amino acids in question have been fixed. Amino acids coloured red are the same in Asibi and ES504, whilst the amino acids coloured orange are the same in 17D and Asibi.

8. Suppl. Fig. 12: The second last processing stage depicted is basically a 20-angstrom ball, and then several steps are abbreviated as '3D refinement, post-processing and Bayesian polishing' and then a 6.5 Å particle is presented. This seems an amazing increase in resolution given the inputs. Can you please present a step-by-step diagram of all steps of this pathway, including FCS curves for all the 3D classified particles, and any intermediaries presented along the way. Again, a clear tyrosine is labelled as K96 in Fig. 12e.
Supplementary Fig. 12 has been updated to include several additional processing steps to help clarify the SPA pathway. Regarding FSC requests for 3D classification steps, Relion 3D classification does not directly output Fourier Shell Correlation (FSC) curves as half-maps are not generated during this process. Additionally, the mislabelled tyrosine has also been amended, we thank the reviewer for picking up this mistake.
9. Suppl. Fig. 21: Could you highlight the differences between the two conformations in Fig. Suppl. 21d? This is not clear from the figure or the text.
A new panel has been added to Supplementary Fig. 21 which highlights the differences between the two confirmations.
10. Suppl. Table 2: Have the information on "other flavivirus vaccines" and "Pre-exposure to other flaviviruses" been used for analysis in this paper? If yes, what are the results?
If not, they can be removed from the table.
The "Other flavivirus vaccines" and "Pre-exposure to other flaviviruses" columns have been removed from Supplementary Table 2.
11. Suppl. Table 4 and 5: Could the authors specify the pixel size at data collection and what type of binning (I assume they bin the data) they applied? Was the binning applied on the images or on the particles?
The pixel size at data collection is already stated in Supplementary Tables 4 and 5 as the "Nominal Pixel Size". The images were binned during processing, this information has been added into the Image Processing and Single-Particle Analysis methods section.
12. To allow for better comparison of 17D:2C9 and ES504-DIII:2C9 (Fig2b and Fig 4a), the same colouring should be used. In Fig2b: coloured by radial distance, Fig4a: coloured by local resolution. Fig 4a: same scale needed for local resolution to compare the particles better.

The bYFV_{17D}:2C9 map is coloured by local resolution in Supplementary Fig. 7g; this should be sufficient to compare the bYFV_{17D}:2C9 (Fig. 2b) and bYFV_{ES504}/DIII:2C9 (Fig. 4c) reconstructions.

The authors believe that if the cryo-EM maps in Fig 4 had the same scale, you would lose a lot of the valuable information regarding the localised resolution of the reconstructions. This is demonstrated below, with the maps shown on the same scale.

13. The ELISA binding data with selected antibodies (Suppl. Fig.4a) show less difference between 17D and wt viruses than the neutralization data presented in Fig.1e. It might be helpful to show a table with the EC50 of the ELISA data. Have the authors considered the effect of temperature on the different viruses tested? The neutralization assays were done at 28C, but the ELISA binding was performed at 37C. Would this affect the “bumpy” structure and the exposure of the FL? Since the reduction of “bumpiness” is proposed as a vaccine approach and vaccines will be acting at 37C, the relation temperature-structure is relevant.

The authors would like to ensure the reviewers that we have considered the effect of temperature on the bYFV virions. As stated in the methods, the neutralisation assays were performed by first incubating mAb and virus together for 1 hour at 37°C. Additionally, the discrepancy between the ELISA binding data and the neutralisation assays was mentioned and critically analysed in paragraph 2 of the discussion.

Reviewer 3:

1. The experiments missing from the paper are structural studies with a fusion loop reactive antibody. The paper presents data with two monoclonal antibodies that neutralize all three strains tested (albeit to varying degrees). The key questions raised by the authors (and interesting potential impact on the vaccine field) relate to access to the fusion loop. While the fusion loop antibodies fail to neutralize YFV17D, how they engage the “bumpy” forms is of considerable interest and relevance. I suspect fusion loop mAbs also bind YF-17D in solution, too. Does mAb binding alter the arrangement of 17D E proteins to look more like Asibi?

We agree with the reviewer that this is an important research question. In response, we have performed complexing and imaging of fusion loop specific Fab (2A10G6) together with 17D and ES504 derived particles (shown in Supplemental Fig. 4). To our knowledge this is the first such study for any mature flavivirus. Full SPA datasets were not collected due to particle aggregation. However, sufficient micrographs of non-aggregated particles were collected to allow qualitative assessment.

Interestingly, we find that the 2A10G6 Fab appears to bind well to ES504 derived particles but only a few 2A10G6 Fab like densities can be seen for 17D preparations (at matched concentration). In contrast, the DIII reactive 17D-specific Fab 864 complexed well with 17D particles but not ES504. We did not proceed to SPA analysis with the fusion loop Fab as it is expected that the particle heterogeneity will prevent refinement to high resolution using icosahedral based approaches (noting that there have been no structures of mature particles solved with a fusion loop antibody bound for any flavivirus, likely due to this same limitation). Nevertheless, this visualisation is important to understand our observed neutralisation profiles in the context of cryo-EM and we thank the reviewer for this suggestion.

2. How do the authors define virion stability (as related to line 428)? An appealing aspect of the paper is the connection between structure data and measures of virion function.

The heterogeneity underlying the bumpy reconstructions may not correspond to virions with reduced infectivity over time (stability).

We thank the reviewer for picking up on this point. We acknowledge that without definition use of stability may be confusing. We do not refer to shelf stability (e.g. half-life) of the particles but are referring to rigidity. Our findings are consistent with a model where the R380 mutation provides additional inter-raft contacts in the “classical” mature flavivirus herringbone arrangement, which lock (or stabilise) this conformation. We have altered some of the discussion to highlight this important point.

Final Responses to reviewer 1 (and 2):

The authors have addressed the majority of our concerns. Few points remain not clearly explained in the “Result” section and only briefly mentioned in the “Discussion”.

For example, the rationale of freezing the viruses at 4C is not explained. A comparison between 4C and 37C for the binding of antibodies before freezing is presented and it seems that, differently from other flaviviruses, there is no correlation between binding and virus breathing. This is barely mentioned in the discussion but not clearly analysed.

Virus samples were frozen at 4 °C as this condition was chosen during vitrification optimisation for reproducibility. Further details have been added into corresponding methods section.

The discrepancy between ELISA and neutralization is not underscored in the “Results” section and justified in the “Discussion” section by the use of detergent in the assay which clearly affect the structure of the virion. Correlation between ELISA on whole virion and neutralization has been well documented for other flaviviruses such for example TBE. If the discrepancy shown in this paper is due to a technical limitation, we wonder what is the relevance of the presented data.

We now provide more context and include a reference to previous work which demonstrates that immobilisation of purified flaviviruses on ELISA plates enhances fusion loop epitope exposure (PMID: 16973559). This prior work shows that, for cryptic epitopes, ELISA and neutralisation data may not be correlated. Furthermore, it is important to present the ELISA data, as it confirms that FLE mAbs can bind to bYFV-17D and therefore lack of neutralisation is not a result of local sequence variation at the epitope, but instead a result of altered epitope exposure between 17D and the virulent strains.

Overall, the data presented in this paper are new and we believe will have an impact on the field.